# Chromosome architecture and low cohesion bias acrocentric chromosomes towards aneuploidy during mammalian meiosis

Eirini Bellou [1], Agata P. Zielinska[1], Eike Urs Mönnich [1], Nina Schweizer [1], Antonio Z. Politi [2], Antonina Wellecke [1], Claus Sibold[3], Andreas Tandler-Schneider[3] & Melina Schuh [1] ✉

Aneuploidy in eggs is a leading cause of miscarriages or viable developmental syndromes. Aneuploidy rates differ between individual chromosomes. For instance, chromosome 21 frequently missegregates, resulting in Down Syndrome. What causes chromosome-specific aneuploidy in meiosis is unclear. Chromosome 21 belongs to the class of acrocentric chromosomes, whose centromeres are located close to the chromosome end, resulting in one long and one short chromosome arm. We demonstrate that acrocentric chromosomes are generally more often aneuploid than metacentric chromosomes in porcine eggs. Kinetochores of acrocentric chromosomes are often partially covered by the short chromosome arm during meiosis I in human and porcine oocytes and orient less efficiently toward the spindle poles. These partially covered kinetochores are more likely to be incorrectly attached to the spindle. Additionally, sister chromatids of acrocentric chromosomes are held together by lower levels of cohesin, making them more vulnerable to age-dependent cohesin loss. Chromosome architecture and low cohesion therefore bias acrocentric chromosomes toward aneuploidy during mammalian meiosis.

Chromosomes often segregate incorrectly during oocyte meiosis[1–3]. Chromosome segregation errors are the most common cause of miscarriages, age-related aneuploidy, and viable developmental syndromes such as Down syndrome[4,5].

Intriguingly, aneuploidy rates differ strongly between chromosomes[3,6,7]. For instance, chromosome 21, whose missegregation is viable and causes Down syndrome, has a very high error rate in both female and male meiosis[4,8]. Recent technical developments in preimplantation genetic diagnosis and SNP genotyping allowed the generation of large datasets on error rates of individual chromosomes. Overall, chromosomes 13, 14, 15, 21, and 22 are more likely to missegregate than all other chromosomes in human spermatocytes[9–11], and chromosomes 15, 16, 21, and 22 missegregate most frequently during female meiosis[3,6,12–17]. Chromosomes 15, 16, 21, and 22 are also the most likely to be aneuploid in miscarried embryos[18] and embryo

biopsies[19–22], and these errors are mainly of maternal origin[23,24]. Aneuploidy in eggs will result in embryonic lethality for most chromosomes, with the exception of chromosome 21 (trisomy causes Down Syndrome), chromosome 13 (trisomy causes Patau Syndrome) and chromosome 18 (trisomy causes Edward Syndrome), and the X-chromosome, where an incorrect number can cause Triple X, Klinefelter Syndrome and Turner Syndrome[5].

Together, these studies suggest a pattern of chromosome segregation errors during human meiosis: the most commonly missegregating chromosomes appear to belong to the class of acrocentric chromosomes, which comprises chromosomes 13, 14, 15, 21, and 22 in humans. In acrocentric chromosomes, the centromere is located very close to the end of the chromosome, giving rise to one long chromosome arm and one very short chromosome arm. The remaining human chromosomes have their centromeres in the central region of the

[1]Department of Meiosis, Max Planck Institute for Multidisciplinary Sciences, Göttingen, Germany. [2]Facility for Light Microscopy, Max Planck Institute for Multidisciplinary Sciences, Göttingen, Germany. [3]Fertility Center Berlin, Berlin, Germany. ✉e-mail: melina.schuh@mpinat.mpg.de

chromosome and thus have two arms of similar size. Chromosomes whose centromeres are close to the centre are referred to as meta-centric, whereas those whose centromeres are slightly off-centre are referred to as sub-metacentric[25]. In this study, we set out to investigate whether acrocentric chromosomes are more likely to missegregate during oocyte meiosis and to identify the mechanisms that could bias them toward aneuploidy.

## Results

### Acrocentric chromosomes missegregate more often than metacentric chromosomes in porcine oocytes

First, we aimed to determine if acrocentric chromosomes mis-segregate more frequently during oocyte meiosis than metacentric chromosomes. Studies in human oocytes are limited by sample scar-city. Although mouse oocytes are commonly used to study meiosis, the female western house mouse (Mus musculus) typically has only telocentric chromosomes, with their kinetochores located at the very end of the chromosome[26,27]. Pigs ovulate multiple oocytes per cycle and have multiple offspring within a litter, whereas humans mostly ovulate one oocyte and have single offspring. Despite this difference, porcine oocyte meiosis resembles human oocyte meiosis[28,29], and like human oocytes, porcine oocytes have both acrocentric and meta-centric chromosomes[30]. In addition, porcine oocytes are easily obtained from abattoirs and are generally of a young age (obtained from gilts), which eliminates potential age-associated differences in error rates between chromosomes[3,31–37]. Young porcine oocytes are expected to already show significant aneuploidy rates[38–41]. Thus, we used porcine oocytes to uncover potential differences between acro-centric and metacentric chromosomes during meiosis.

We developed a strategy to differentially label acrocentric and metacentric chromosomes in live porcine oocytes, based on targeting fluorescent protein fusions to arrays of repeated DNA sequences[42,43]. Pericentromeric regions consist of large arrays of DNA repeats; importantly, in porcine cells, all acrocentric chromosomes share a similar pericentromeric sequence that is absent from metacentric chromosomes. Furthermore, several metacentric chromosomes share a pericentromeric sequence that is absent from all acrocentric chro-mosomes and the rest of the metacentric chromosomes[44]. These fea-tures allowed us to design two fluorescently labelled Transcription Activator Like Effectors (TALEs) that selectively label the two chro-mosome classes (Fig. 1a, b).

Injecting porcine oocytes with the TALE for the acrocentric chro-mosomes (acrocentric-TALE) resulted in a labelled group of acrocentric chromosomes ("acro-labelled") and an unlabelled group of metacentric chromosomes ("meta") (Supplementary Fig. 1a). Expression of the acrocentric-TALE led to a readily detectable signal in the pericen-tromeric regions of all six acrocentric chromosomes in porcine oocytes, which was present throughout oocyte maturation (Supplementary Fig. 1b, c; Supplementary Movie 1). As expected, the metacentric-TALE labelled 6 out of the 13 metacentric chromosomes ("meta-labelled"), and was also readily detectable throughout oocyte maturation (Sup-plementary Fig. 1b, c; Supplementary Movie 2). FISH probes targeting the same loci resulted in the same pattern, verifying the specificity of the TALEs (Supplementary Fig. 1d). The metacentric-TALE was used to ensure that phenotypes did not arise from labelling per se. The unla-belled chromosomes in the metacentric-TALE-expressing oocytes comprise both acrocentric and metacentric chromosomes (Supple-mentary Fig. 1a, middle panel), and were not quantified.

We first examined the aneuploidy rates by kinetochore counting in fixed metaphase II porcine eggs (Fig. 1c). A deviation in chromosome number at metaphase II would indicate an error during anaphase of meiosis I. Oocytes were injected with the acrocentric-TALE after completing meiosis I, thus the anaphase outcome was not influenced by the presence of the TALE. In total, 26% of the eggs were aneuploid (Fig. 1d). This is in line with aneuploidy levels reported previously for

eggs from gilts[39–41]. Acrocentric chromosomes were significantly more likely to be affected by aneuploidy (7%) than metacentric chromo-somes (1%) (Fig. 1e and Supplementary Table 1). We also followed acro- and meta-TALE labelled chromosomes in live oocytes from late metaphase I until metaphase II. Oocytes were injected with the acro- or metacentric-TALE in the GV stage during meiosis I. We found that 23% of the porcine oocytes failed to segregate the chromosomes correctly during anaphase, and had at least one chromosome segregate to the wrong pole, resulting in an aneuploid egg (Fig. 1f). The missegregation frequency for the acro-labelled chromosomes was 8%, significantly higher than the missegregation frequencies for the meta chromo-somes (1%) and meta-labelled chromosomes (0%) (Fig. 1g). Overall, these data demonstrate that acrocentric chromosomes are more fre-quently aneuploid than metacentric chromosomes in porcine oocytes, consistent with high aneuploidy rates reported for acrocentric chro-mosomes in human meiosis[1,3].

### Acrocentric chromosomes are more likely to misalign during late metaphase I and to lag during anaphase

A recent study showed that during mitosis, aneuploidy rates of individual chromosomes are affected by their location in the nucleus[7], with chro-mosomes in the periphery of the nucleus more likely to missegregate than chromosomes in the central region of the nucleus. We examined the location of the meta- and acrocentric chromosome-specific TALEs in the porcine oocyte nucleus and found that similar to mitosis of human cells[7,45], acrocentric chromosomes were more centrally located than metacentric chromosomes (Supplementary Fig. 2a). The higher error rates of acrocentric chromosomes during meiosis are thus not due to a more peripheral localisation of this chromosome group in the nucleus. Furthermore, porcine and human oocytes cluster all chromosomes in the nucleus centre before spindle assembly begins[46]. The original posi-tion of the different chromosomes in the GV nucleus is therefore likely to be less important in female meiosis compared to mitosis.

Aneuploidy is often caused by chromosomes that fail to align on the metaphase plate or that lag behind during anaphase[34–36,47,48]. To examine the behaviour of acrocentric and metacentric chromosomes during chromosome alignment and segregation, we co-expressed the acrocentric-TALE or the metacentric-TALE with the kinetochore mar-ker mScarlet-CENPC. We imaged chromosome congression at high spatiotemporal resolution using light sheet microscopy. We devel-oped a script that automatically tracks all kinetochores in 3D and assigns kinetochore pairs to paired homologous chromosomes, called a bivalent. Kinetochore assignments and tracking were executed simultaneously by utilising bivalent intensities and orientations, which enhances the accuracy of the tracking and reduces the need for manual corrections of the tracks (Fig. 2a).

Compared to metacentric chromosomes, acrocentric chromo-somes were less likely to align on the metaphase plate by the time of anaphase onset. We found that acrocentric chromosomes were on average located around ~1 μm further away from the metaphase plate than metacentric chromosomes (Fig. 2b, c and Supplementary Fig. 2b, c). Both small and large acrocentric chromosomes were loca-ted away from the metaphase plate, showing that the chromosomes' misalignment was independent of chromosome size (Supplemen-tary Fig. 2d).

We then examined chromosome segregation behaviour during anaphase I by following acro- and meta-labelled chromosomes in live oocytes. We found that 33% of segregation errors in porcine oocytes were due to chromosomes that failed to align on the metaphase plate by the time of anaphase onset, and 66% of segregation errors were due to chromosomes that lagged behind the main chromosome masses during anaphase (Fig. 2d). Next, we scored the lagging chromosomes as mild (still lagging at 12 min after anaphase onset) or severe (still lagging at 20 min after anaphase onset) (Fig. 2e, Supplementary Movies 3–5). Compared to metacentric chromosomes, acro-labelled chromosomes

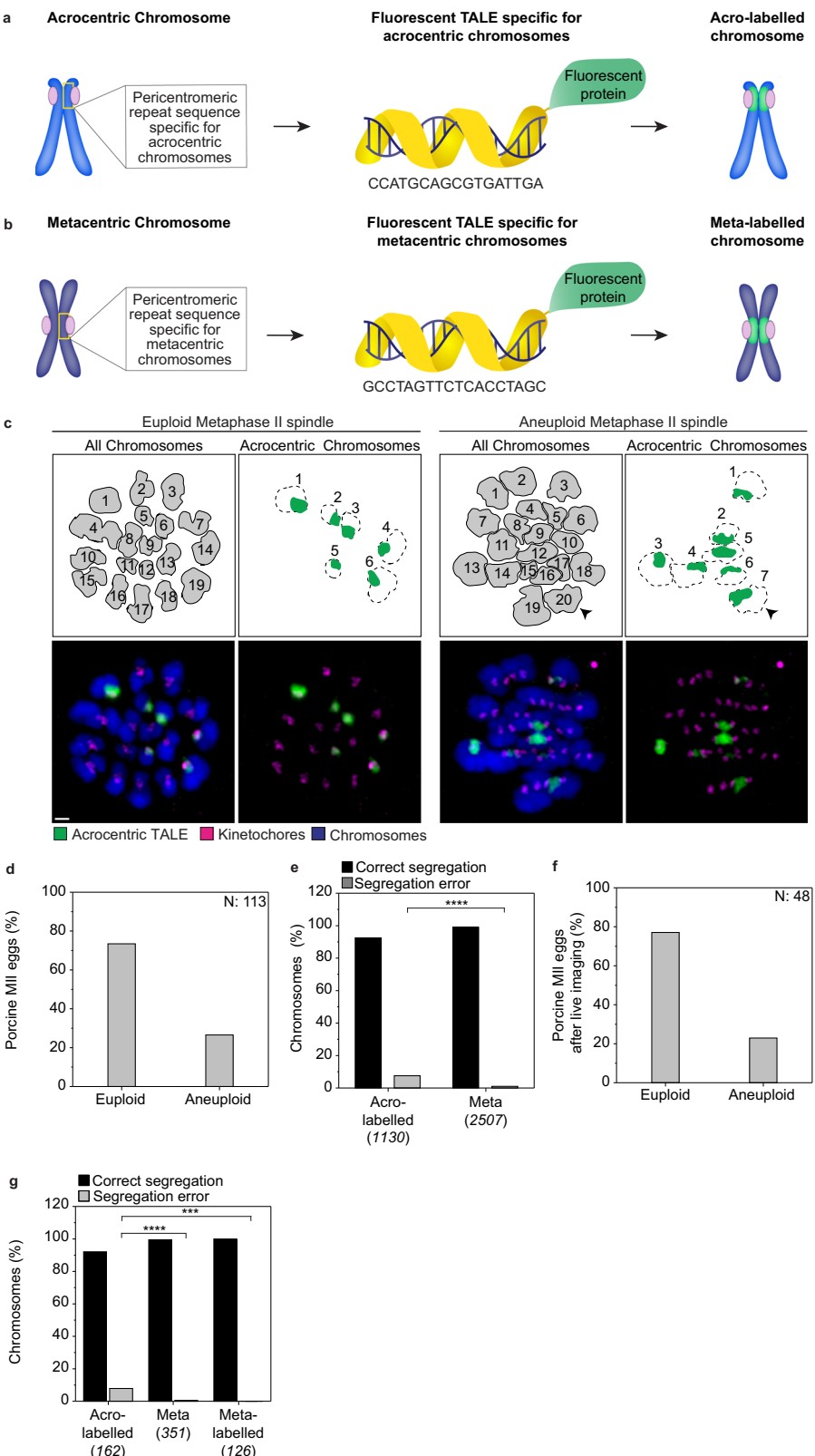

were four times more likely to display mild lagging, and six times more likely to display severe lagging (Fig. 2f, g). Meta-labelled chromosomes behaved similarly to the non-labelled metacentric chromosomes, indicating that the labelling per se did not affect lagging (Fig. 2f, g). Overall, our data indicate that acrocentric chromosomes are significantly more likely to misalign during metaphase and lag behind during anaphase compared to metacentric chromosomes.

## Porcine acrocentric chromosomes frequently form erroneous kinetochore-microtubule attachments

Misaligned and lagging chromosomes are typically caused by the formation of erroneous kinetochore-microtubule attachments, such as merotelically attached kinetochores, which are attached to both spindle poles, and laterally attached kinetochores, which attach to the sides rather than the ends of microtubules[47,49–55]. We, therefore,

**Fig. 1 | Acrocentric chromosomes missegregate more often than metacentric chromosomes in porcine oocytes. a, b** Schematic representation of the labelling strategy with the pericentromeric TALEs in the porcine cells. The region recognised by each TALE is indicated for each chromosome type. TALEs are fused with a fluorescent protein. Labelled chromosomes are referred to as acro-labelled and meta-labelled, respectively. Blue, chromosomes; magenta, kinetochores; yellow, TALE; green, fluorescent protein, and TALE on chromosomes. **c** Representative immunofluorescence images of a spindle from a euploid and an aneuploid intact fixed metaphase II egg injected with the acrocentric-TALE, (bottom row). Numbers within the grey illustrations indicate the number of chromosomes in each of the two spindles (top row). Numbers next to the green illustrations indicate the number of acro-labelled chromosomes in both the euploid and aneuploid cells. The black arrowheads indicate the extra acro-labelled chromosome. Scale bar, 1 μm. Representative examples of 96 immunolabelled porcine eggs. **d** The fraction of aneuploid eggs from the total of analysed fixed metaphase II porcine eggs. Aneuploid eggs were identified by kinetochore counting (113 eggs analysed). **e** Frequency of missegregation for acro-labelled and meta chromosomes from the total chromosomes counted in each category. Counting was performed in fixed metaphase II porcine spindles injected with the acrocentric-TALE (96 eggs analysed). Two-sided Fisher's exact test ($p < 0.0001$). **f** Proportion of euploid and aneuploid metaphase II porcine eggs after live-cell imaging. Euploidy status is determined by the correct or erroneous segregation outcome of the chromosomes during live imaging (48 eggs analysed). **g** Frequency of missegregation per chromosome group after live-cell imaging (48 eggs analysed). Two-sided Fisher's exact test (acro-labelled/meta, $p < 0.0001$; acro-labelled/meta-labelled, $p = 0.003$). The number of chromosomes analysed is indicated in brackets under each category. P-values in the graphs are indicated as follows, ***$p < 0.001$ and ****$p < 0.0001$. The number of cells is indicated with $N$ in the graph or in brackets in the figure legend.

analysed whether acro- and metacentric chromosomes differ in how they attach to spindle microtubules during meiosis I, when homologous chromosomes segregate, and meiosis II when sister chromatids segregate. Notably, the sister kinetochores of one homologue must attach to the same spindle pole in metaphase I, whereas in metaphase II the sister chromatids attach to opposite spindle poles, similar to their orientation on the spindle in mitosis[1,4] (Fig. 3a).

We examined cold-treated metaphase I porcine spindles (to selectively visualise k-fibres) during late spindle assembly (12 h post-release) and classified the microtubule attachments to the sister kinetochores into three categories: end-on (correct attachment), lateral/merotelic (incorrect attachment), and unattached (Fig. 3b). We found that about 50% of sister kinetochores of acro-labelled chromosomes formed end-on attachments compared to about 64–75% of paired kinetochores of meta and meta-labelled chromosomes (Fig. 3c). Furthermore, over 34% of sister kinetochores of acro-labelled chromosomes formed lateral/merotelic attachments in meiosis I compared to less than 18% of sister kinetochores of meta and meta-labelled chromosomes (Fig. 3c). In addition, the number of unattached kinetochores was similar in all groups. This was further verified by Mad1 quantifications, which is a marker for unattached kinetochores. We found that the number of Mad1-positive kinetochores for all chromosomes, during late spindle assembly (also at 12 h post-release), was 29% (Supplementary Fig. 3a, b). This result is in line with the high fraction of unattached kinetochores that was observed in the cold-stable assay.

Next, we analysed the kinetochore-microtubule attachments in cold-treated metaphase II porcine spindles (Fig. 3b – Metaphase II). Here, most kinetochores formed end-on attachments to spindle microtubules and, importantly, we did not observe a difference in attachment types between acro- and metacentric chromosomes in metaphase II porcine eggs (Fig. 3d). Overall, these data indicate that acrocentric chromosomes are more likely to be incorrectly attached to spindle microtubules than metacentric chromosomes during meiosis I, but not during meiosis II.

## Kinetochores that are partially masked by the small chromosome arm are more likely to be incorrectly attached to microtubules

The chiasmata that link the two homologous chromosomes within a bivalent during meiosis I differ in number and location for meta- and acrocentric chromosomes. Metacentric chromosomes usually form chiasmata on both of their arms[9,56,57], and therefore typically have two chiasmata, whereas acrocentric chromosomes are generally thought to form chiasmata only on their long arm[58,59] (Fig. 4a). We hypothesised that these differences in chiasmata positions for meta- and acrocentric chromosomes are associated with differential accessibility of the sister-kinetochore pairs during meiosis I. For instance, the presence of chiasmata on both arms of metacentric chromosomes will link the arms on each side of the sister-kinetochore pair together and could

thereby help to expose it, facilitating the interaction with spindle microtubules. In contrast, the small arm of acrocentric chromosomes might hinder access to the kinetochores as they protrude from the kinetochore region, preventing microtubule access.

We first investigated where the small arm of acrocentric bivalents is located relative to the kinetochore. Every chromosome end is protected by the telomere complex, so we used the telomeres as a proxy for the position of the small arm on acrocentric chromosomes (Fig. 4a). All acro-labelled chromosomes had telomeres in the proximity of their kinetochores, in contrast to meta and meta-labelled chromosomes (Supplementary Fig. 4a). We observed two types of orientations of the short chromosome arms relative to kinetochores in acrocentric chromosomes in meiosis I: the short arms either partially covered the kinetochores ("telomere-masked kinetochores"; 20% of acrocentric kinetochore pairs; Supplementary Fig. 4b), or they were located toward the centre of the bivalent, leaving the kinetochores largely exposed ("exposed kinetochores"; 80% of acrocentric kinetochore pairs; Supplementary Fig. 4b) (Fig. 4b; Supplementary Fig. 4c, Supplementary Movies 6–14). The telomeres similarly partially covered Hec1 in the outer kinetochore layers of acrocentric chromosomes (Supplementary Fig. 4d).

To investigate whether the telomere-masked and exposed kinetochores differed in their microtubule attachments, we examined attachment types in cold-treated metaphase I porcine spindles (Fig. 4c). We found that 52% of the exposed kinetochores were end-on attached, but only 7% of telomere-masked kinetochores were attached end-on (Fig. 4d). The remaining 93% of telomere-masked kinetochores were either unattached or laterally/merotelically attached in roughly equal fractions. Exposed kinetochores were significantly less likely to be unattached or laterally/merotelically attached than telomere-masked kinetochores (Fig. 4d). We conclude that the partial coverage of kinetochores by telomeres and their associated short chromosome arm correlates with incorrect kinetochore-microtubule attachments.

We investigated if the fraction of telomere-masked versus exposed kinetochores changes during meiosis, by reanalyzing the data in Fig. 4d and Supplementary Fig. 4b. We defined metaphases with broad, unfocused spindle poles and incomplete chromosome alignment as "early metaphase" and those with focused spindle poles and complete chromosome alignment as "late metaphase". The fraction of telomere-masked sister kinetochores decreased from 26% in early metaphase to 14% in late metaphase (Fig. 4e), suggesting that kinetochores can progress from a partially covered to an uncovered state as oocyte meiosis proceeds.

We next asked if the sister kinetochores of acro- and metacentric chromosomes have distinct angles to the spindle axis. To measure the angle between the sister kinetochores and the spindle axis, we performed isotropic 3D-STED imaging[60] followed by automated segmentation of the kinetochore signal (Fig. 4f, g). The angle of each sister-kinetochore pair was determined as the dot-product between

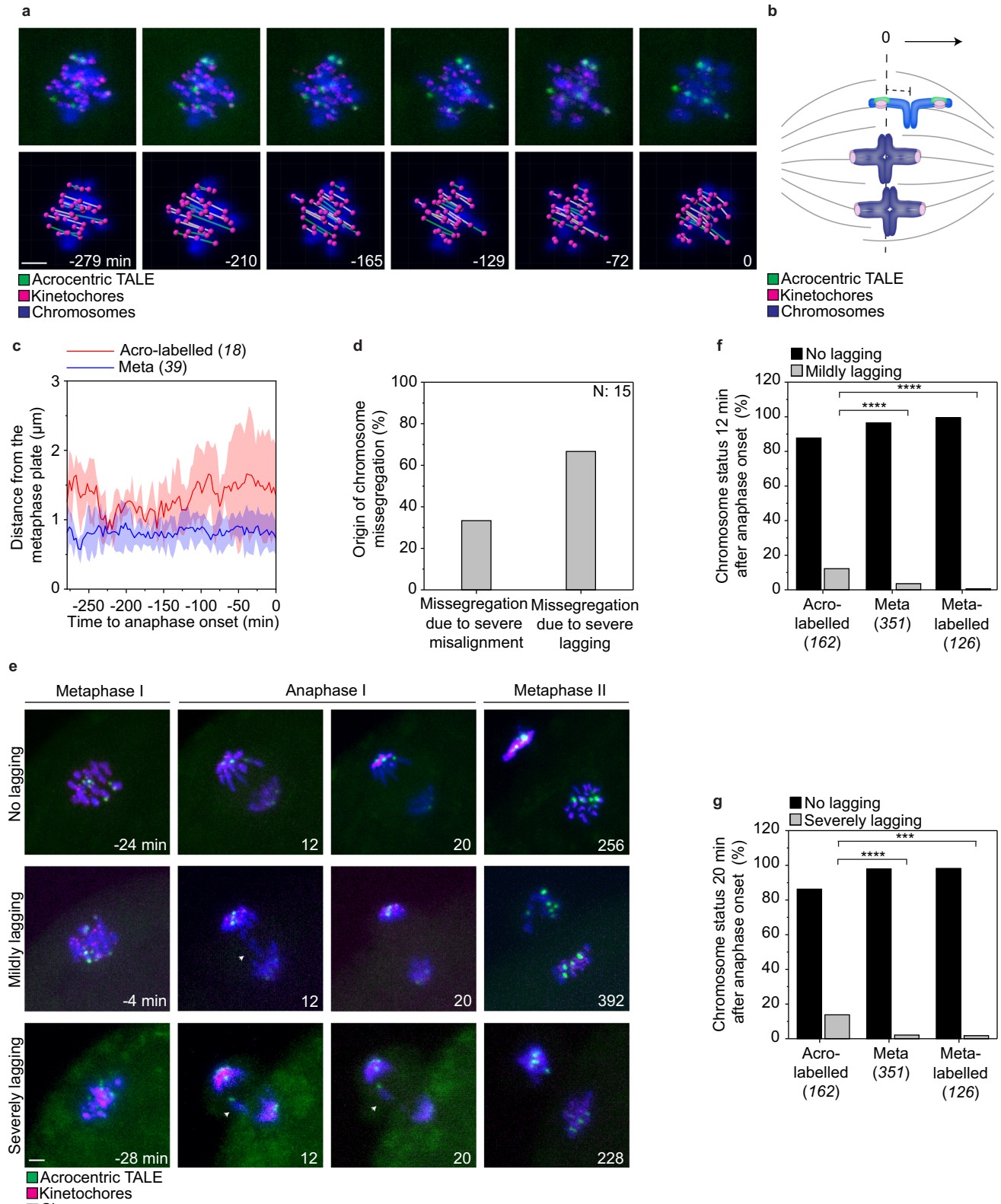

the longest kinetochore axis and the spindle axis. Kinetochore pairs that are oriented perpendicular to the spindle axis are directly facing the spindle poles, whereas those that are oriented parallel to the spindle axis are facing away from the poles, in the direction of the metaphase plate (Fig. 4h, Supplementary Fig. 4e). We found that kinetochore pairs of acro-labelled chromosomes were significantly more parallel to the spindle axis than those of meta and meta-labelled

chromosomes (Fig. 4i, j). A large fraction of acro-labelled kinetochore pairs were either parallel or oriented at around 45 degrees relative to the spindle axis. In contrast, meta and the meta-labelled chromosomes were typically close to a perpendicular position relative to the spindle axis, and unlikely to be oriented parallel (Fig. 4j). Thus, sister-kinetochore pairs of acrocentric chromosomes orient less efficiently to the spindle poles than metacentric chromosomes.

**Fig. 2 | Acrocentric chromosomes are more likely to misalign during late metaphase and lag during anaphase. a** Still images from time-lapse movies of porcine oocytes during chromosome alignment at the metaphase plate (top row) and the corresponding image from Imaris with the pairing annotation (bottom row). Time is indicated in minutes and time point 0 is the last frame before anaphase onset. Magenta, kinetochores (mScarlet-hCENPC); green, acrocentric label, (Acrocentric-TALE-GFP); blue, chromosomes, (H2B-SNAPf) in the top row and magenta, kinetochores; green line, pair of homologous acro-labelled kinetochores; white line, pair of homologous meta kinetochores. Scale bar, 1 μm. **b** Schematic representation shows measurements of a bivalent's distance from the metaphase plate. The Acro-labelled chromosome is outside the centre of the metaphase plate and meta chromosomes align in the middle of the metaphase plate. Magenta, kinetochores; green, acrocentric-TALE; blue, chromosomes; grey, spindle. **c** Quantifications of the distance of acro-labelled and meta bivalents from the centre of the metaphase plate. Error bars (shaded areas) represent SD (3 oocytes analysed). **d** Proportion of the different types of segregation errors from the chromosomes that clearly missegregated during live-cell imaging of anaphase. In the category "Missegregation due to severe misalignment" are chromosome pairs that were severely misaligned at anaphase onset and remained at this position

during anaphase, and hence both chromosomes of the pair were located either in the egg or the polar body; in the category "Missegregation due to severe lagging" are chromosome pairs that severely lagged behind during anaphase and eventually failed to segregate, with both chromosomes again located together in the egg or the polar body (48 oocytes analysed). **e** Still images from a time-lapse movie of chromosome segregation in porcine oocytes. Images from metaphase I, anaphase I, and metaphase II. Time is indicated in minutes and time point 0 is the last frame before anaphase onset. Arrowheads indicate lagging chromosomes. Magenta, kinetochores (mScarlet-hCENPC); green, acrocentric label, (Acrocentric-TALE-GFP); blue, chromosomes, (H2B-SNAPf). Scale bar, 1 μm. **f** Frequency of mildly lagging chromosomes (12 min after anaphase onset) from the total chromosomes examined (48 oocytes analysed). Two-sided Fisher's exact test (acro-labelled/meta, $p < 0.0001$; acro-labelled/meta-labelled, $p < 0.0001$). **g** Frequency of severely lagging chromosomes (20 min after anaphase onset) from total chromosomes examined (48 oocytes analysed). Two-sided Fisher's exact test (acro-labelled/meta, $p < 0.0001$; acro-labelled/meta-labelled, $p = 0.0003$). Number of chromosomes analysed is indicated in brackets next to each category or with $N$ inside the graph. P-values in the graphs are indicated as follows, ***$p < 0.001$ and ****$p < 0.0001$. The number of cells is indicated in brackets in the figure legend.

## Human acrocentric chromosomes often form incorrect kinetochore-microtubule attachments

We next asked whether the kinetochores of acrocentric chromosomes in human oocytes are also partially masked and more likely to be incorrectly attached to microtubules. We identified acro- and metacentric chromosomes based on the position of telomeres relative to kinetochores (Fig. 5a). As expected, we detected bivalents in human oocytes with kinetochores with proximal telomeres (acrocentric) and without proximal telomeres (metacentric) (Fig. 5b). Analysis of kinetochore-microtubule attachments in human oocytes showed that about 50% of kinetochores of acrocentric chromosomes formed end-on attachments compared to about 70% of kinetochores of metacentric chromosomes (Fig. 5c, d), similar to our observations in porcine oocytes. In addition, 23% of kinetochores of acrocentric chromosomes formed lateral/merotelic attachments compared to 13% of kinetochores of metacentric chromosomes (Fig. 5c, d). All human oocytes were monitored for chromosome alignment and fixed at around 15 h 30 min post-NEBD. However, we cannot rule out that individual oocytes might have been at slightly different stages of development at the time of fixation. Thus, also in human oocytes, kinetochores with proximal telomeres were more likely to be incorrectly attached to microtubules than those without proximal telomeres.

We also found that telomeres sometimes partially masked sister-kinetochore pairs on acrocentric chromosomes in human oocytes, as we observed in porcine oocytes. In human oocytes, 38% of kinetochore pairs in acrocentric chromosomes were telomere-masked and 62% were exposed (Fig. 5e, f). Telomere-masked and exposed kinetochores were observed across all ages in human oocytes (Supplementary Fig. 5a). As in porcine oocytes, 30% of telomere-masked kinetochore pairs in human oocytes were end-on attached, and the remaining 70% were either unattached or laterally/merotelically attached. In contrast, about 70% of exposed kinetochore pairs were end-on attached and 30% were either unattached or laterally/merotelically attached (Fig. 5g). Together, these data indicate that the short chromosome arm often partially masks kinetochore pairs during meiosis I in human oocytes, which correlates with incorrect kinetochore-microtubule attachments.

## Pig acrocentric chromosomes have less cohesin than metacentric chromosomes

Acrocentric chromosomes are susceptible to age-related aneuploidy[3,19,61]. The major mechanism involved in age-related aneuploidy is the premature separation of sister chromatids (PSSC)[3,62] due to loss of pericentromeric cohesion[3,35-37,63-65]. We therefore asked whether acro- and metacentric chromosomes differ in their levels of pericentromeric cohesin. To test this, we first measured the

interkinetochore distance between the sister chromatids in porcine eggs fixed at metaphase II. We found significantly larger interkinetochore distances in acrocentric chromosomes (1.09 μm ± 0.18) compared to meta and meta-labelled chromosomes (0.96 μm ± 0.13 and 0.97 μm ± 0.11) (Fig. 6a, b). Furthermore, interkinetochore distances varied more in the acrocentric group of chromosomes than in the metacentric group (Fig. 6b).

Interkinetochore distances correlate with cohesin levels in the pericentromeric region[35,36]. To investigate whether acrocentric chromosomes have lower levels of cohesin, we stained the eggs for SMC3, a universal cohesin subunit (Fig. 6c), and measured the intensity of SMC3 between the two sister kinetochores, as explained in the "Methods" section. Acro-labelled chromosomes had significantly less cohesin than meta and meta-labelled chromosomes (Fig. 6d). In addition, we measured the levels of pericentromeric cohesin on bivalents in meiosis I by staining with an antibody against REC8, a meiosis-specific subunit of the cohesin complex[63]. Intensities were measured within identical spheres around the centre of each kinetochore (Fig. 6e). As in meiosis II, acrocentric chromosomes had significantly lower cohesin levels than metacentric chromosomes (Fig. 6f).

To test for potential differences in the susceptibility of acro- and metacentric chromosomes to PSSC, we acutely removed cohesin during meiosis II by Trim-Away of REC8[28,66,67], and compared the speed of sister chromatid separation between the two groups within the same egg. Chromosomes with lower cohesin levels would be expected to separate first after acute cohesin depletion. The first chromosome to separate after REC8 depletion was acrocentric in 9 out of 10 eggs (Fig. 6g). We further compared the time of separation for each of the six acro-labelled and meta-labelled chromosomes. Acro-labelled chromosomes were generally faster than meta-labelled chromosomes during each separation event (Fig. 6h, i). The high variability in separation time within each group could indicate additional chromosome-specific differences in PSSC susceptibility (Fig. 6h). Together, our data indicate that acrocentric chromosomes have on average higher interkinetochore distances and less cohesin than metacentric chromosomes. However, there is variability between individual chromosomes within each group, consistent with the variability in PSSC rates for individual chromosomes in human oocytes[3,6,12].

## Discussion

Aneuploidy in human oocytes is a leading cause of miscarriages and infertility. Aneuploidy rates vary for individual chromosomes, but the origins of this variability are unknown. In this study, we report that acrocentric chromosomes are more likely to be aneuploid than metacentric chromosomes in porcine eggs. The higher aneuploidy

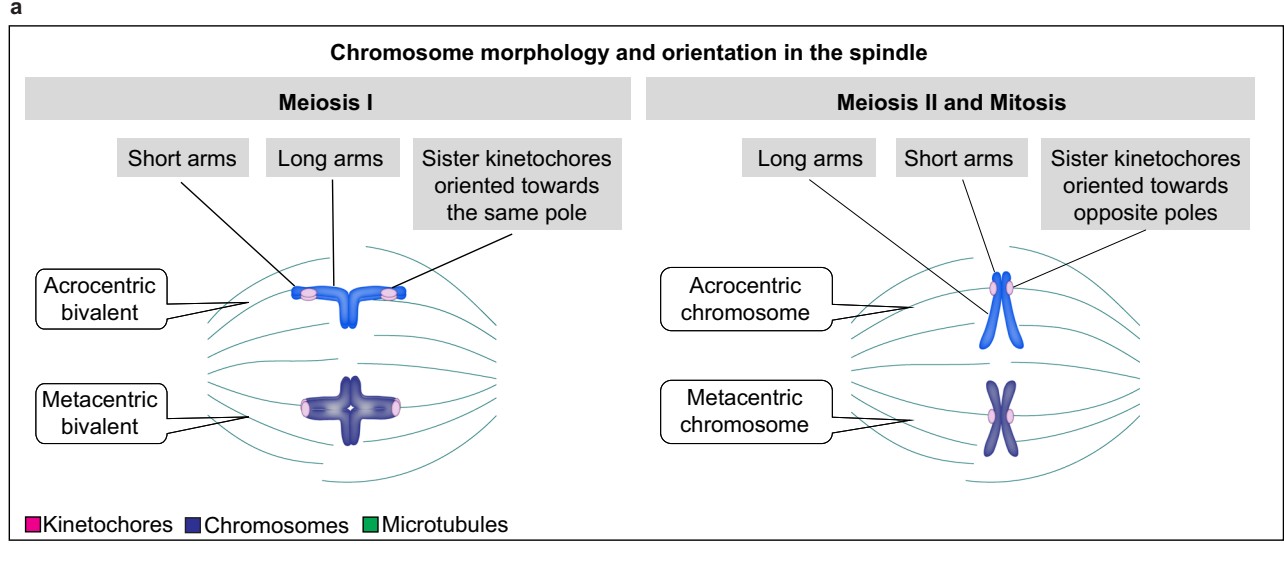

**a** Chromosome morphology and orientation in the spindle

Meiosis I — Acrocentric bivalent, Metacentric bivalent; Short arms, Long arms, Sister kinetochores oriented towards the same pole

Meiosis II and Mitosis — Acrocentric chromosome, Metacentric chromosome; Long arms, Short arms, Sister kinetochores oriented towards opposite poles

Kinetochores, Chromosomes, Microtubules

**b** Porcine oocytes

Metaphase I: End-on, Lateral/Merotelic, Unattached; Metaphase II: End-on

Acro-labelled, Meta, Meta-labelled

Acrocentric - TALE, Kinetochores, Microtubules, Chromosomes

Metacentric - TALE, Kinetochores, Microtubules, Chromosomes

**c** Acro-labelled (150), Meta (348), Meta-labelled (94). Kt-Mt attachments in MI (%). End-on **, *; Lateral/Merotelic ***, ***; Unattached n.s., n.s.

**d** Acro-labelled (124), Meta (372), Meta-labelled (133). Kt-Mt attachments in MII (%). End-on n.s.; Lateral/Merotelic n.s.; Unattached n.s.

rates of acrocentric chromosomes appear to be conserved in human meiosis when sequencing-based studies of chromosome aneuploidy rates are assessed cumulatively[3,6,9–17]. A microscopy-based assessment of aneuploidy rates of acrocentric chromosomes is not currently possible in human metaphase II eggs, but may become possible in the future with further technical development and could then complement the sequencing-based data.

We report that acrocentric bivalents align less efficiently than metacentric bivalents on the first meiotic spindle of porcine oocytes, with their sister kinetochores generally more angled away from the

**Fig. 3 | Porcine acrocentric chromosomes frequently form erroneous kinetochore-microtubule attachments. a** Schematic illustration of the chromosome morphologies and the orientation in the spindle. Examples of acrocentric and metacentric chromosomes in meiosis I and meiosis II/mitosis. The small arm of acrocentric chromosomes and the orientation of the sister kinetochores are indicated in both cases. **b** Illustrations and representative immunofluorescence images of kinetochore-microtubule attachments from cold-treated porcine oocytes in Metaphase I (left side) and Metaphase II (right-side). Overview images of a z-plane of the full spindle (top row) and insets of the attachments (bottom row). Insets are magnifications of regions marked by dashed line boxes in the overview image. Magenta, kinetochores, (ACA); grey, TALE, (anti-GFP); blue, chromosome, (Hoechst); green microtubules (anti-a-tubulin). Scale bar, 1 μm for both overview and insets. **c** Quantification of the proportion of kinetochore-microtubule (Kt-Mt) attachments for acro-labelled, meta, and meta-labelled chromosomes in meiosis I spindles in porcine oocytes. The number of attachments quantified per chromosome type is indicated in brackets (25 oocytes analysed). Two-sided Fisher's

exact test (end-on: acro-labelled/meta, $p = 0.0216$; end-on: acro-labelled/meta-labelled, $p = 0.007$; latera/merotelic: acro-labelled/meta, $p = 0.0002$; latera/merotelic: acro-labelled/meta-labelled, $p = 0.0002$; unattached: acro-labelled/meta, $p = 0.1783$; unattached: acro-labelled/meta-labelled, $p = 0.4426$). **d** Quantification of the proportion of kinetochore-microtubule (Kt-Mt) attachments for acro-labelled, meta, and meta-labelled chromosomes in meiosis II spindles in porcine oocytes. The number of attachments quantified per chromosome type is indicated in brackets (26 cells analysed). Two-sided Fisher's exact test (end-on: acro-labelled/meta, $p = 0.8101$; end-on: acro-labelled/meta-labelled, $p = 1.000$; latera/merotelic: acro-labelled/meta, $p = 0.6862$; latera/merotelic: acro-labelled/meta-labelled, $p = 0.6230$; unattached: acro-labelled/meta, $p = 1.0000$; unattached: acro-labelled/meta-labelled, $p = 0.4332$). Number of kinetochores analysed is indicated in brackets. *P*-values in the graphs are indicated as follows, $*p < 0.05$, $**p < 0.01$, $***p < 0.001$ and n.s.: non-significant. The number of cells is indicated in brackets in the figure legend.

---

spindle poles than sister kinetochores of metacentric chromosomes. Moreover, acrocentric chromosomes are more likely than metacentric bivalents to lag behind during anaphase, leading to chromosome segregation errors and higher aneuploidy rates during metaphase II.

Accurate chromosome alignment and segregation are critically dependent on the formation of correct kinetochore-microtubule attachments. Acrocentric chromosomes are frequently incorrectly attached to microtubules in porcine and human oocytes. Incorrect attachments are particularly prominent in bivalents where the short arm of acrocentric chromosomes partially covers the kinetochore region. Our data suggest that specifically the formation of end-on attachments during the late stages of meiosis is delayed for acrocentric chromosomes, whereas lateral attachments are still efficiently formed. This is consistent with a model in which the short arm masks parts of the kinetochore, making it more difficult for acrocentric chromosomes to form end-on kinetochore-microtubule attachments. In turn, the formation of end-on attachments could help to expose kinetochores and contribute to the increase in exposed kinetochores as meiosis proceeds. The unique architecture of acrocentric chromosomes could also hinder the formation of end-on microtubule attachments through additional mechanisms. It is likely that during meiosis I, kinetochores of acrocentric chromosomes will be under less tension compared to metacentric chromosomes where both arms have undergone meiotic recombination. This may affect the efficiency of spindle assembly checkpoint signalling. We report that kinetochores of acrocentric chromosomes are less likely to be oriented toward the spindle poles during meiosis I compared to those of metacentric chromosomes, which is likely to result in an altered distribution of forces on the kinetochore surface. In addition, acrocentric chromosomes on average have higher interkinetochore distances and lower pericentromeric cohesin levels, which likely favours PSSC[31,34–36,68,69]. Our findings thus provide several explanations for why acrocentric chromosomes are frequently aneuploid in human eggs (Fig. 7a).

Acrocentric chromosomes are also more likely than metacentric chromosomes to missegregate during human spermatogenesis[9–11]. Also during male meiosis, chiasmata are thought to form only on the long chromosome arm of acrocentric chromosomes[58,59]. The short chromosome arms are therefore also in close proximity to the kinetochores during male meiosis I. Kinetochore masking may thus also contribute to the high error rates of acrocentric chromosomes during male meiosis.

Metacentric chromosomes typically form chiasmata on both chromosome arms[57]. Chiasmata are required to link homologous chromosomes together[4], but one chiasma would be sufficient to achieve homologous chromosome pairing. Why metacentric chromosomes form two chiasmata has been unclear. Our results suggest that kinetochore exposure during meiosis I is important for the efficient formation of end-on microtubule attachments and accurate chromosome

segregation. The formation of chiasmata on both chromosome arms may help to keep kinetochores exposed and therefore well accessible during meiosis I, providing a possible explanation for the formation of a chiasma on each arm of metacentric chromosomes (Fig. 7b).

While acrocentric chromosomes are likely to be affected by aneuploidy during meiosis, they are less likely to be aneuploid than metacentric chromosomes during mitosis[7,70–73]. We propose that this difference can be explained by differences in the architecture of acrocentric chromosomes during meiosis I and mitosis: during mitosis, sister kinetochores are oriented toward opposite spindle poles, and the sister chromatids are therefore oriented perpendicular to the spindle axis. The short arm will not partially mask the kinetochore in this orientation, allowing unobstructed access to spindle microtubules (see "Results" section). Consistent with this model, acrocentric chromosomes were equally likely to be correctly attached to microtubules during meiosis II (see "Results" section) when chromosomes adopt an orientation on the spindle that is similar to their orientation in mitosis.

Chromosome-specific error rates during female meiosis are also affected by ageing[1,3], which causes a release of cohesin from chromosomes, resulting in premature dissociation of homologous chromosomes and sister chromatids[35–37,63]. We report that acrocentric chromosomes have less pericentromeric cohesin than metacentric chromosomes in porcine oocytes and eggs. It is expected that this causes them to be more vulnerable to PSSC after age-related cohesin loss. It remains to be elucidated if cohesin levels and the rate of cohesion loss vary between chromosome groups in human oocytes and how this contributes to the error patterns observed in human female meiosis. For instance, small chromosomes and chromosomes with unfavourable chiasmata positions are likely to be linked by lower levels of cohesin in the arm region during meiosis I, and may hence also be more likely to dissociate prematurely during meiosis I upon cohesin loss in aged oocytes[3,74]. Chromosome size should not affect PSSC rates though, as PSSC is caused by weakening of cohesion in the pericentromeric region. Currently, there are limited data on individual chromosome error rates during meiosis I and II, due to the low availability of human oocytes and generally a small number of data points for every chromosome. The generation of expanded datasets will be helpful for increasing our understanding of the causes of chromosome-specific error rates, also in the context of ageing.

Species differ in their chromosome morphologies. Unlike porcine cells, which have a human-like karyotype, bovine cells do not have metacentric chromosomes but mainly acrocentric chromosomes[75,76]. Interestingly, bovine oocytes have a higher rate of aneuploidy than porcine oocytes[29,38,39,77]. In contrast, mice have only telocentric chromosomes and therefore lack a short arm that could otherwise mask the kinetochore. Oocytes from young mice (3 months old) are very unlikely to be aneuploid (4.1% of all oocytes)[68] and have much lower aneuploidy rates than human and porcine oocytes. The absence of the

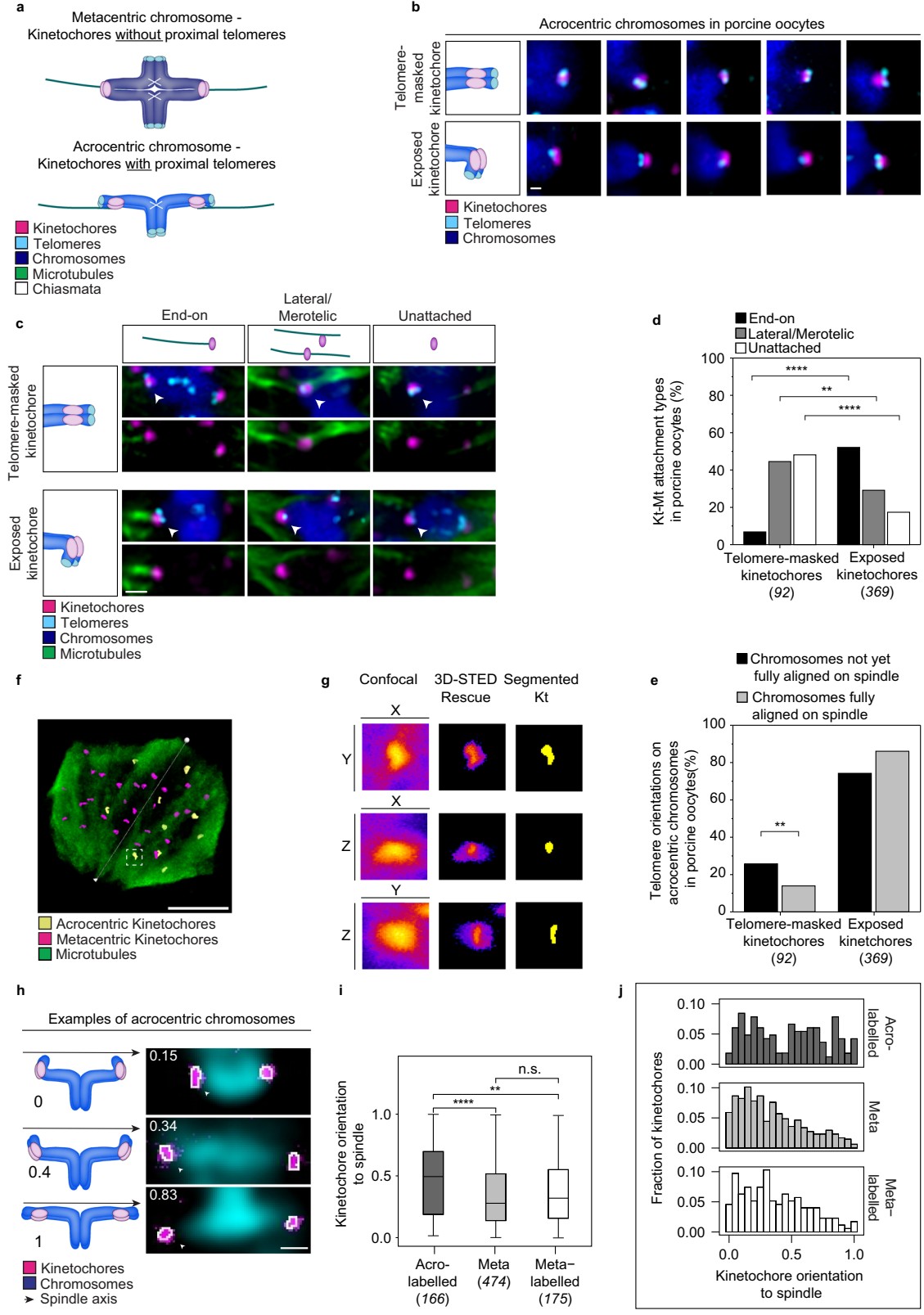

unique architectural features that bias acrocentric bivalents toward aneuploidy may therefore contribute to the low aneuploidy rates of mouse oocytes. Furthermore, the length of the short arm may vary between acrocentric chromosomes of different species, which in turn may influence the level of interference.

Taken together, our data reveal striking differences in how chromosome-specific aneuploidy rates arise in mitosis and meiosis.

During mitosis, different features of chromosomes have been linked to chromosome-specific aneuploidy rates[71,78], including the location of the chromosomes in the nucleus, which determines whether chromosomes have difficulty entering the central region of the spindle and subsequently fail to align by the time of anaphase onset[7]. Interestingly, human and porcine oocytes transiently form a compact cluster of chromosomes, which brings all chromosomes into close proximity[46].

**Fig. 4 | Kinetochores partially masked by the small chromosome arm are more likely to be incorrectly attached to microtubules. a** Illustrations show differences in the telomere's position with respect to the kinetochore in acrocentric and metacentric bivalents when aligned in a metaphase plate. Differences in the lengths of the arms result in differences in the distance of the telomeres from the kinetochores in the two morphologies. Magenta, kinetochores; cyan, telomeres; blue, chromosomes; green, spindle axis; white, chiasmata. **b** Illustrations and representative Airyscan immunofluorescence images with telomere-masked kinetochores (top row) and exposed kinetochores (bottom row) in porcine oocytes. Magenta, kinetochores, (ACA); cyan, telomeres, (TRF-2); blue, chromosomes, (Hoechst). Scale bar, 0.5 μm. Representative examples of 44 immunolabelled porcine oocytes. **c** Illustrations and representative Airyscan immunofluorescence images from the attachment types of telomere-masked and exposed kinetochores in porcine oocytes. Arrowheads indicate the specified attachment type. Magenta, kinetochores, (ACA); cyan, telomeres, (TRF-2); blue, chromosomes, (Hoechst); green, microtubules, (a-tubulin). Scale bar, 1 μm. Representative examples of 44 immunolabelled porcine oocytes. **d** Quantification of the proportion of kinetochore-microtubule (Kt-Mt) attachments for telomere-masked and exposed kinetochores of acrocentric chromosomes in porcine oocytes (44 oocytes analysed). Two-sided Fisher's exact test (end-on, $p < 0.0001$; latera/merotelic, $p = 0.0087$; unattached, $p < 0.0001$). **e** Distribution of telomere-masked and exposed kinetochores of acrocentric chromosomes on spindles with fully aligned chromosomes and spindles that have not yet aligned their chromosomes (44

oocytes analysed). Two-sided Fisher's exact test ($p = 0.0016$). **f** Rendering of segmented acro-labelled kinetochores (yellow, TALE labelled) and non-labelled kinetochores (magenta) in a meiotic metaphase spindle (microtubules in green). The white arrow indicates the spindle direction. Scale bar, 5 μm. Representative examples of 32 immunolabelled porcine oocytes. **g** Magnification of the kinetochore shown in F, white square, and the corresponding confocal, 3D-STED, and segmented kinetochores in XY, XZ, and YZ views. Each tile is 2 × 2 μm. **h** Illustrations and representative images of different angles of kinetochores in acrocentric chromosomes. The black arrow on the left side represents the spindle axis. The numbers in each image are the dot products between the left kinetochore longest axis and spindle axis. Magenta, kinetochores; blue, chromosomes; black, spindle axis. **i** Box plot showing the kinetochore orientation to the spindle for the three kinetochore populations (32 oocytes analysed). Box plot shows the median (horizontal black line), mean (small black squares), 25th and 75th percentile (boxes), and outliers (whiskers). Wilcoxon two-tailed test, (acro-labelled/meta, $p = 2.38e - 06$; acro-labelled/meta-labelled, $p = 0.002$; meta/meta-labelled, $p = 0.181$). **j** Distribution of the dot-product for the kinetochore orientation to the spindle for acro-labelled (upper graph), meta (middle graph), and meta-labelled kinetochores (lower graph) (32 oocytes analysed). Number of kinetochores analysed is indicated in brackets next to each category. P-values in the graphs are indicated as follows, $**p < 0.01$, $****p < 0.0001$, and n.s.: non-significant. The number of cells is indicated in brackets in the figure legend.

The spindle subsequently forms from within the chromosome aggregate[79]. Chromosomes from peripheral positions in the nucleus may therefore be less likely to be affected by aneuploidy in oocytes compared to mitotic cells. Meiotic aneuploidy rates are instead related to chromosome-specific cohesin levels and the architecture of the bivalent. Age-related cohesin loss is specific to oocytes, and bivalents do not form during mitosis, which may explain why acrocentric chromosomes are prone to high aneuploidy in meiosis but not in mitosis.

## Methods
### Preparation and culture of porcine oocytes
Porcine ovaries from gilts were obtained from a local slaughterhouse and transported to the laboratory in a thermo-flask. For each experimental repetition, oocytes were collected from at least 5 different animals. Oocyte isolation and culture were performed using collection and maturation media from Cosmobio. In brief, the dissection of antral follicles from the surface of porcine ovaries was performed in a homemade M2 medium supplemented with 10 μM RO-3306 (#SML0569) at 38.5 °C. The follicular fluid was transferred to a 50 ml tube. Cumulus oocyte complexes (COCs) were allowed to sediment and then washed extensively with an M2 medium. In the final wash, COCs were transferred to porcine oocyte/embryo collection medium (POE-CM, Cosmobio, #CK020) supplemented with 10 μM RO-3306. All handling and manipulations outside the $CO_2$ incubator were performed in a POE-CM medium. Only oocytes with a homogeneous cytoplasm and surrounded by at least 3–5 complete layers of compact cumulus cells were selected for experiments. The selected oocytes were transferred to a basic medium for porcine oocyte maturation (POM, Cosmobio, #CK021) supplemented with 10 IU/ml equine chorionic gonadotropin (eCG), 10 IU/ml human chorionic gonadotropin (hCG), and 10 μM RO-3306 for meiotic arrest. COCs were incubated in the media at 38.5 °C in a 5% $CO_2$ incubator. After 6 h, COCs were partially denuded using an EZ-Grip denudation pipettor with a 135 μm tip. All handlings outside the $CO_2$ incubator were performed in the POE-CM apart from live-cell imaging which was performed in POM. To induce the resumption of meiosis, oocytes were washed in an RO-3306-free medium. Complete denudation of the oocytes was performed before release from the RO-3306-induced arrest. For complete imaging of Meiosis I or for imaging of the chromosomes in the GV nucleus, oocytes were transferred to the light-sheet sample chamber immediately after release. For imaging of anaphase I, oocytes were transferred to the light-sheet chamber or the confocal imaging dish

11 h after release. For immunofluorescence staining of metaphase I spindles and for FISH experiments, oocytes were fixed, at late MI, 12 h after release. Oocytes that required >40 h to complete maturation were excluded from all analyses, as longer maturation times have been linked to abnormal oocyte development[80].

### Expression constructs and mRNA synthesis
For the generation of the constructs for mRNA synthesis, previously published coding sequences were fused with mClover3[81] or mScarlet[82]. Acrocentric-TALE and Metacentric-TALE plasmids were generated by fusing together the repeat variable diresidues (RVDs) recognising the DNA sequence CCATGCAGCGTGATTGA for acrocentric chromosomes and GCCTAGTTCTCACCTAGC for metacentric chromosomes[44] in a pTALYM3 vector, described in more detail below. Acrocentric-TALE was subsequently inserted in a pGEMHE-mClover and a pGEHME-mScarlet vector using pENTR/D-TOPO Gateway cloning (Invitrogen). pGEMHE-mScarlet-hCENPC[83], pGEMHE-mCherry-TRIM21[66], pGEHME-mScarlet-Mad1 (gift from Katarina Harasimov), pGEMHE-H2B-SNAPf and pGEMHE-3xEGFP-TRF1 (gift from Jan Ellenberg) were also used. All mRNAs in this study were synthesised using HiScribe T7 ARCA mRNA kit (NEB #E2065S) according to the manufacturer's protocol and quantified using Qubit RNA HS Assay Kit (Thermo Fisher Scientific #Q32852).

### Creation of fluorescent TALES
Fluorescent TALEs were created according to a previously published protocol[84]. RVD plasmids were obtained from Addgene (Kit #1000000024). TALEs were assembled by Golden Gate cloning in TOP10 competent cells (Thermo Fisher Scientific; C404010). The assembled TALEs were further subcloned into the destination vectors pGEMHE-mclover3-N1 and pGEMHE-mScarlet-N1 using pENTR/D-TOPO cloning (Invitrogen).

### Microinjection of porcine oocytes
Porcine oocytes were microinjected with 2.5 pl of mRNAs, as described previously[85,86] and also below, in a microinjection chamber created with two layers of double-sided sticky tape between two parallel coverslips. The microinjection system used in this study utilises small droplets of mercury to regulate the pressure and ensure precise volumes of mRNA, in the picolitre range, are injected into the oocytes. Microinjection was performed after the partial denudation of oocytes. mRNAs were injected at the following concentrations on the

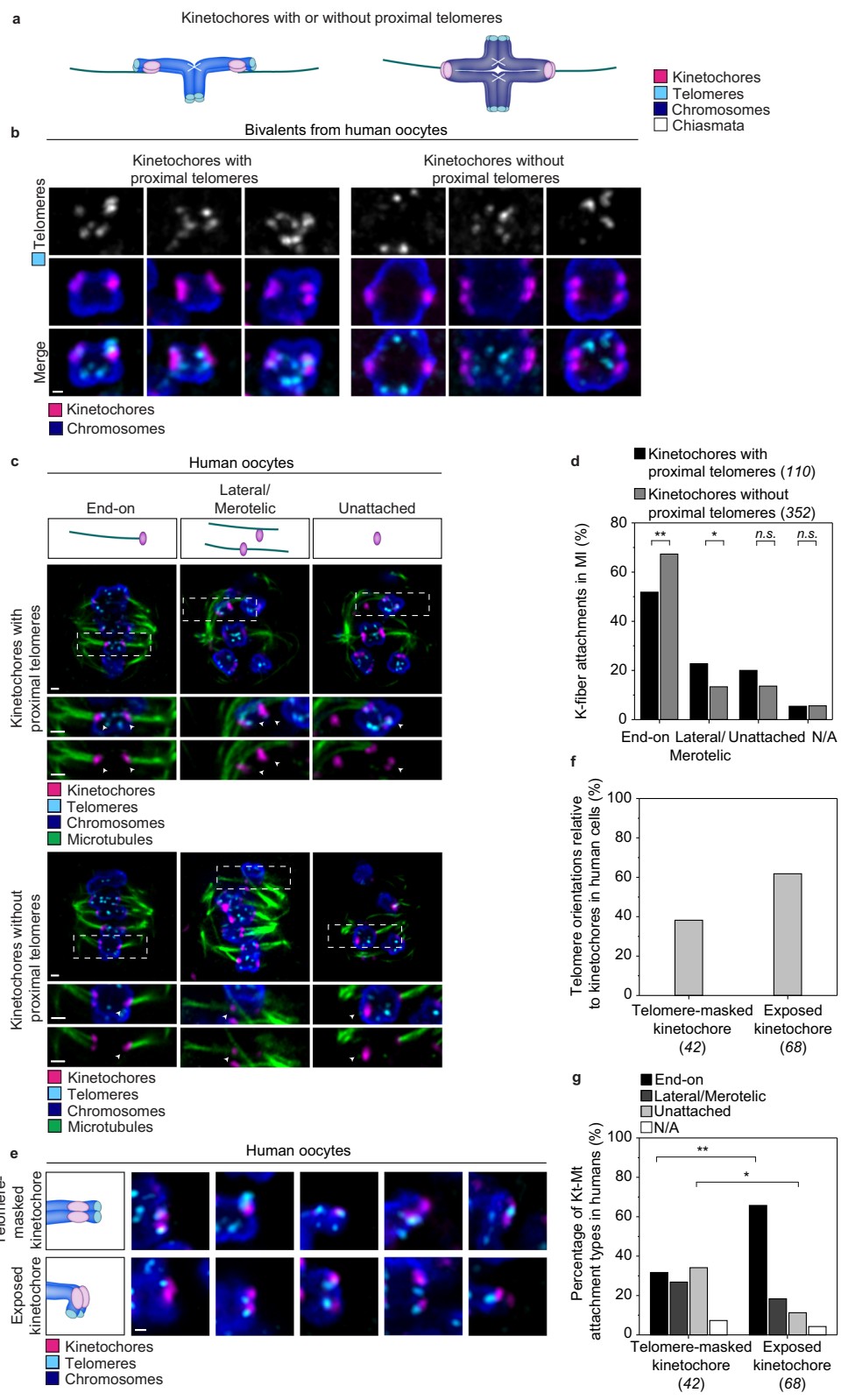

microinjection needle: H2B-SNAPf at 9 ng/µl, mScarlet-hCENPC at 10 ng/µl, Acrocentric-TALE at 50 ng/µl, Metacentric-TALE at 50 ng/µl, mcherry-TRIM21 at 150 ng/µl, mScarlet-Mad1 at 30 ng/µl. After injection oocytes were allowed to express the mRNAs in a medium supplemented with RO-3306 for 3 h before release. For experiments in Fig. 1d, e mRNAs were injected into metaphase II eggs. Eggs were allowed to be expressed for 5 h before fixation.

## Trim-Away of REC8 in MII porcine oocytes

REC8 antibody used for Trim-Away was generated in the lab using a previously characterised epitope[87]. The antibody was concentrated using Ultra 0.5 mL Amicon 100kD centrifugal filters (Merck #UFCS510024). The concentrated antibody was supplemented with NP40 (Merck #492016) to a final concentration of 0.05% and centrifuged for 14.000 × g for 10 min before injection.

**Fig. 5 | Human acrocentric chromosomes frequently form erroneous kinetochore-microtubule attachments. a** Schematic representation of bivalents kinetochores with telomeres in proximity and with kinetochores without telomere in proximity. Chromosomes in blue; kinetochores in magenta; telomeres in cyan; chiasmata in white. **b** Representative examples of immunofluorescence Airyscan images of human bivalents with kinetochores with telomeres in proximity and kinetochores without telomeres in proximity. Bivalent chromosomes are expected to have 8 telomeres, but in some cases, subsets of telomeres are in close proximity of each other so that fewer than 8 spots are detected Magenta, kinetochores, (ACA); cyan, telomeres, (TRF-2); blue, chromosome, (pH3). Scale bar, 0.5 μm. Representative examples of chromosomes from 10 immunolabelled human oocytes. **c** Illustrations and representative immunofluorescence images of kinetochore-microtubule attachments from cold-treated human oocytes in meiosis I. Overview images of a z-plane of the spindle (top row) and insets of the attachments at the bottom rows. Insets are magnifications of regions marked by dashed line boxes in the overview image. Arrowheads indicate the specified attachment type. Magenta, kinetochores, (ACA); cyan, telomeres, (TRF-2); blue, chromosome, (pH3); green microtubules, (a-tubulin). Scale bar, 1 μm for both overview and insets.

**d** Quantification of the proportion of the different kinetochore-microtubule attachments in kinetochores with telomeres in proximity and kinetochores without telomere in proximity (10 oocytes analysed). Two-sided Fisher's exact test (end-on, $p = 0.0044$; lateral/merotelic, $p = 0.0235$; unattached, $p = 0.1268$; N/A, $p = 1.000$). **e** Illustrations and representative immunofluorescence images with telomere-masked kinetochores (top row) and exposed kinetochores (bottom row) from human oocytes. Magenta, kinetochores, (ACA); cyan, telomeres, (TRF-2); blue, chromosomes, (Hoechst). Scale bar 0.5 μm. **f** Proportion of telomere-masked and exposed kinetochores in acrocentric chromosomes from the total kinetochores of acrocentric chromosomes examined (10 oocytes analysed). **g** Quantification of the proportion of the different kinetochore-microtubule (Kt-Mt) attachment types for telomere-masked and exposed kinetochores of acrocentric chromosomes in human oocytes (10 oocytes analysed). Two-sided Fisher's exact test (end-on, $p = 0.0017$; unattached, $p = 0.0160$). Number of chromosomes or kinetochores analysed is indicated in brackets under each category. $P$-values in the graphs are indicated as follows, $*p < 0.05$, $**p < 0.01$, $***p < 0.001$, and n.s.: non-significant. The number of cells is indicated in brackets in the figure legend.

Trim-Away was performed by injecting mRNA for Trim21 at a final concentration of 150 ng/μl at the germinal vesicle (GV) stage together with other mRNAs (H2B-SNAPf and the two TALEs). Oocytes were released after 3 h of expression and allowed to mature till metaphase II. Only oocytes with a polar body were selected for antibody injection. 34 h after release MII eggs were injected with the REC8 antibody. Only a few oocytes were injected at a time and immediately transferred to the light-sheet sample chamber for imaging. Imaging was performed at one-minute intervals until sister chromosome separation occurred.

## Human oocyte culture and injection

The use of unfertilised human oocytes in this study was approved by the Ärztekammer Niedersachsen (Ethics Committee of Lower Saxony) under reference 15/2016. Oocytes were collected from patients who underwent ovarian stimulation for intracytoplasmic sperm injection (ICSI) as part of their assisted reproduction treatment at Fertility Center Berlin. Only oocytes that were immature at the time of ICSI and thus unsuitable for the procedure were vitrified for this study with Cryolock (FUJIFILM Irvine Scientific) using Vit Kit-Freeze (FUJIFILM Irvine Scientific). All patients gave informed consent for their surplus oocyte(s) to be used in this study. Oocytes were thawed according to previously published protocol[29] also described below. Specifically, oocytes were thawed in 1 ml of prewarmed G-MOPS PLUS (Vitrolife) containing 1 M d-(+)-trehalose (Sigma-Aldrich) at 37 °C for 1 min. Then, oocytes were transferred to 300 μl of G-MOPS PLUS containing 0.5 M d-(+)-trehalose at room temperature for 3 min, 300 μl of G-MOPS PLUS containing 0.25 M d-(+)-trehalose at room temperature for 5 min, and 300 μl of G-MOPS PLUS at room temperature for 2 min. After that oocytes were cultured in G-MOPS medium (Vitrolife, #10129) supplemented with 10% FBS (GIBCO, #16000044) under mineral oil (Nidoil, Nidacon #NO-400K) at 37 °C[28]. Oocytes were injected with 3xEGFP-TRF1 mRNA at 200 ng/μl immediately after thawing. To assess the maturation stage and monitor the chromosome alignment, the medium was supplemented with 10 nM 5-SiR-Hoechst DNA[88] for chromosome staining[29]. At late MI, about 15 h 30 min post-NEBD and when chromosomes were largely aligned at the metaphase plate, as monitored under the microscope, oocytes were cold treated and fixed as described below.

## Immunofluorescence

To obtain porcine oocytes with metaphase I and metaphase II spindles, oocytes were fixed at 12 h and 24 h after release from RO-3306. Oocytes were matured in a 39 °C/ 5% CO$_2$ incubator before fixation. To obtain human metaphase I spindles, thawed human Meiosis I oocytes were monitored every 30 min on a confocal LSM880 microscope at 38.5 °C as described below.

Both porcine and human oocytes were fixed in 100 mM HEPES (pH 7.0, titrated with KOH), 50 mM EGTA (pH 7.0, titrated with KOH), 10 mM MgSO4, 2% methanol-free formaldehyde and 0.5% triton X-100 (Sigma-Aldrich, 93443) at room temperature for 30 min. Fixed oocytes were extracted in phosphate-buffered saline (PBS) with 0.5% triton X-100 (PBT) overnight at 4 °C and then blocked in PBT with 5% BSA (Fisher Scientific #BP1605) (PBT-BSA) overnight at 4 °C. Lipid droplets in porcine oocytes were cleared with 4000 U/ml lipase from *Candida rugose* (Sigma-Aldrich) in 400 mM NaCl, 50 mM Tris (pH 7.2), 5 mM CaCl$_2$, and 0.2% sodium taurocholate supplemented with cOmplete, EDTA-free Protease Inhibitor Cocktail (Roche) at room temperature for 20–40 min[89] before staining with primary antibodies. All primary antibody incubations were performed overnight at 4 °C in PBT-BSA at the concentrations listed below. Secondary antibodies and Hoechst 33342 incubations were performed in PBT-BSA for 1 h at room temperature.

Primary antibodies used were human anti-centromere antibody (ACA) at 1:50 dilution (Antibodies Incorporated #15-234), rabbit anti-GFP (#A11122, Invitrogen), rat anti-alpha-tubulin (MCA78G, Bio-rad), mouse anti-TRF-2 (NB100-56506SS), goat anti-GFP (600-101-215; Rockland Immunochemicals), mouse anti-alpha-tubulin (#T6199, Merck), rabbit anti-pH3 (#9701, Cell Signaling Technology), rabbit anti-REC8 generated in-house based on a published epitope 87, rabbit anti-SMC3 (#ab128919, abcam), mouse anti-HEC1 (#ab3613, Abcam) at 1:100 dilution. Secondary antibodies used in this study were Goat anti-Human IgG (H+L) Cross-Adsorbed Secondary Antibody, Alexa Fluor 647 (#A-21445, Thermo Fisher), Goat anti-Human IgG (H+L) Cross-Adsorbed Secondary Antibody, Alexa Fluor 488 (#A-11013, Thermo Fisher), Donkey anti-Rabbit IgG (H+L) Highly Cross-Adsorbed Secondary Antibody, Alexa Fluor 488 (#A-21206, Thermo Fisher), Goat anti-Rabbit IgG (H+L) Highly Cross-Adsorbed Secondary Antibody, Alexa Fluor 568 (#A-11036, Thermo Fisher), Donkey anti-Mouse IgG (H+L) Highly Cross-Adsorbed Secondary Antibody, Alexa Fluor 568 (#A-10037, Thermo Fisher), Donkey anti-Mouse IgG (H+L) Highly Cross-Adsorbed Secondary Antibody, Alexa Fluor 488 (#A-21202, Thermo Fisher), Goat anti-Rat IgG (H+L) Cross-Adsorbed Secondary Antibody, Alexa Fluor 488 (#A-11006, Thermo Fisher), Goat anti-Rat IgG (H+L) Cross-Adsorbed Secondary Antibody, Alexa Fluor 568 (#A-11077, Thermo Fisher), Donkey anti-Goat IgG (H+L) Cross-Adsorbed Secondary Antibody, Alexa Fluor 568 (#A-11057, Thermo Fisher), Donkey Anti-Rabbit IgG H&L Alexa Fluor 405 preadsorbed (#ab175649, Abcam), Rhodamine Red-X (RRX) AffiniPure Fab Fragment Goat Anti-Mouse IgG2a, Fcγ fragment specific (#115-297-186, Jackson ImmunoResearch), Alexa Fluor 647 AffiniPure Fab Fragment Goat Anti-Mouse IgG1, Fcγ fragment specific (#115-607-185, Jackson ImmunoResearch), Donkey anti-human STAR RED (#STRED-1054-500UG, Abberior Gmbh), Goat anti-rabbit STAR ORANGE (#STORANGE-1002-

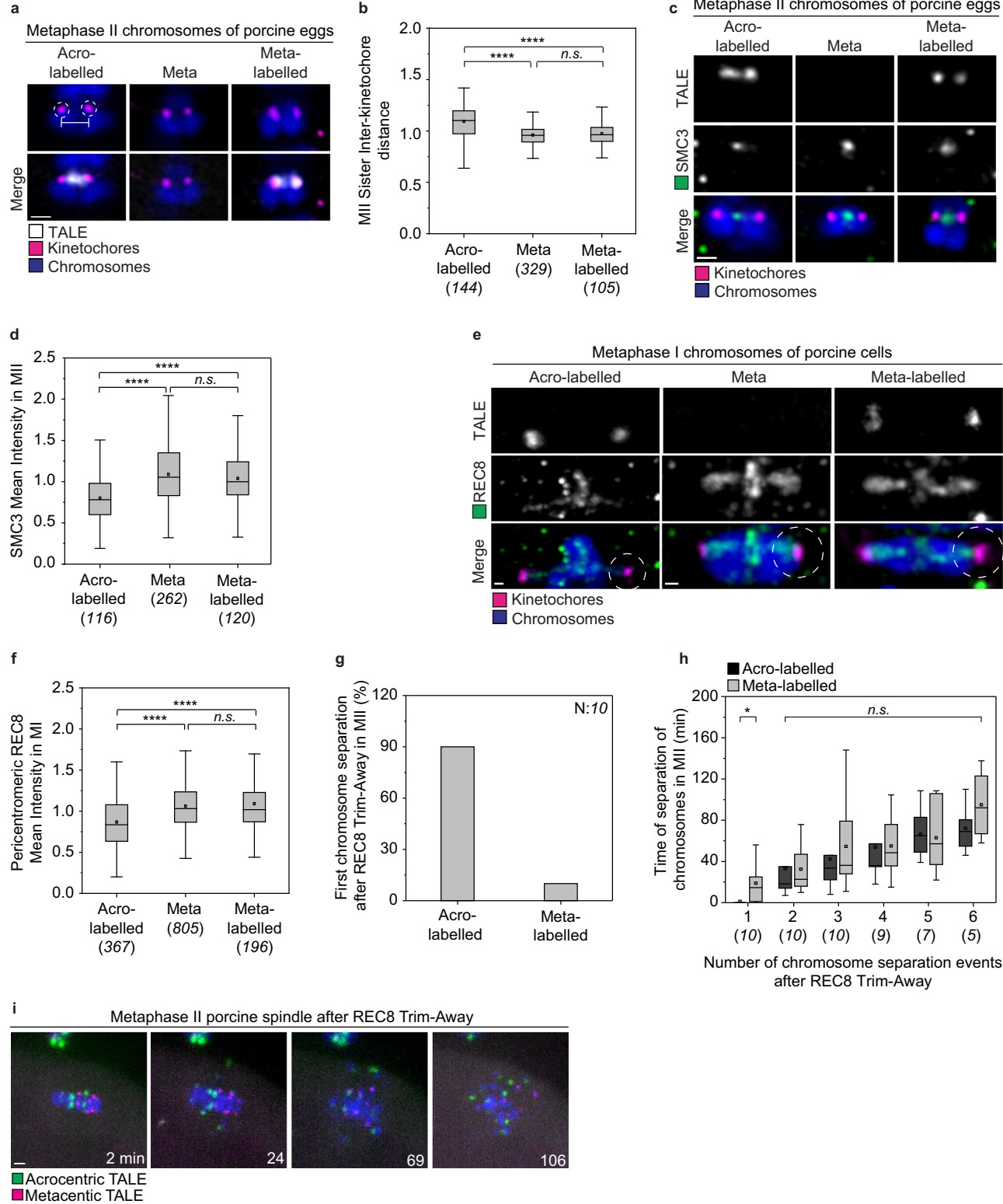

500UG, Abberior Gmbh) all at 1:200 dilution. Hoechst 33342 was used at 100 μM to stain the DNA (Molecular Probes; 20 mM stock).

### Cold-stable assay

Porcine metaphase I and II oocytes were incubated on ice for 9 min, while human metaphase I and metaphase II oocytes were incubated on ice for 5.5 min. Subsequently, they were fixed for immunofluorescence staining as described above.

### Confocal and super-resolution microscopy

For live confocal imaging, oocytes were placed in 5 μl of M2 medium under mineral oil (Light mineral oil, Irvine Scientific #9305) in a 35 mm dish with a #1.0 coverslip (MATTEK, #P35G-1.0-1.4-C). Just before imaging, oocytes injected with H2B-SNAPf were incubated with 2 μM SNAP-cell 637-SiR (NEB, S9102S) supplemented with 10 μM Verapamil (Spirochrome; #SC007) for 1 h. Images were acquired on an LSM880 confocal laser scanning microscope (Zeiss) equipped with an

**Fig. 6 | Pig acrocentric chromosomes have less cohesin than metacentric chromosomes. a** Representative Airyscan immunofluorescence images of the measurements from the sister chromatids in the three different chromosome groups. Magenta, kinetochores, (ACA); grey, TALE (anti-GFP); blue, chromosomes, (Hoechst); Dashed circles indicate the sister kinetochores and the white line indicates the distance measured. Scale bar, 1 µm. **b** Box plot showing measurements of the interkinetochore distance of sister chromatids on metaphase II intact spindles of porcine oocytes (41 oocytes analysed). Box plot shows the median (horizontal black line), mean (small black squares), 25th and 75th percentile (boxes), and outliers (whiskers). Two-tailed unpaired *t*-test was performed (acro-labelled/meta, $p = 1.41075e - 8$; acro-labelled/meta-labelled, $p = 7.51899e - 9$; meta/meta-labelled, $p = 0.31898$). **c** Representative Airyscan immunofluorescence images of the SMC3 staining on chromosomes in metaphase II intact spindles of porcine eggs. Single z-plane images from the three different chromosome groups. Magenta, kinetochores, (ACA); grey, TALE (anti-GFP); blue, chromosomes, (Hoechst); green, cohesin, (anti-SMC3); Scale bar, 1 µm. **d** Box plot showing mean intensity measurements of SMC3 signal on chromosomes from metaphase II intact spindles of porcine eggs (40 oocytes analysed). Box plot shows the median (horizontal black line), mean (small black squares), 25th and 75th percentile (boxes), and outliers (whiskers). Two-tailed unpaired *t*-test was performed, (acro-labelled/meta, $p = 6.11772e - 7$; acro-labelled/meta-labelled, $p = 2.04894e - 4$; meta/meta-labelled, $p = 0.37062$). **e** Representative Airyscan immunofluorescence images of the REC8 staining on chromosomes from metaphase I intact spindles of porcine oocytes. Images from the three different chromosome groups. Images are projections of 16 z-planes. Dashed circles indicate the area of the measurements.

Magenta, kinetochores, (ACA); grey, TALE (anti-GFP); blue, chromosomes, (Hoechst); green, cohesion, (anti-REC8); Scale bars, 0.5 µm. **f** Box plot of mean intensity measurements of REC8 intensity around the pericentromeric areas on chromosomes from metaphase I intact spindles of porcine oocytes (49 oocytes analysed). Box plot shows the median (horizontal black line), mean (small black squares), 25th and 75th percentile (boxes), and outliers (whiskers). Two-tailed unpaired *t*-test was performed (acro-labelled/meta, $p = 3.89493e - 24$; acro-labelled/meta-labelled, $p = 1.41981e - 13$; meta/meta-labelled, $p = 0.20548$). **g** Percentage of the first chromosome to separate after Trim-Away of REC8 in metaphase II porcine eggs. **h** Time of separation for each of the six acro-labelled and meta-labelled chromosomes after Trim-Away of REC8 in porcine eggs. As time point zero in each egg is the time of the first chromosome separation (10 oocytes analysed). Box plot shows the median (horizontal black line), mean (small black squares), 25th and 75th percentile (boxes), and outliers (whiskers). Two-tailed unpaired *t*-test was performed between each separation event for acro-labelled and meta-labelled (1st separation, $p = 0.01326$; 2nd separation, $p = 0.96525$; 3rd separation, $p = 0.47748$; 4th separation, $p = 0.94742$, 5th separation, $p = 0.82915$, 6th separation, $p = 0.2905$). **i** Still images from a time-lapse movie after Trim-Away of REC8 in metaphase II porcine eggs. Time is indicated in minutes and time point 0 is the first frame when imaging started after the injection of the REC8 antibody. Magenta, Metacentric-TALE (Metacentric-TALE-GFP); green, acrocentric label, (Acrocentric-TALE-mScarlet); blue, chromosomes, (H2B-SNAPf). Scale bar, 1 µm. Number of chromosomes analysed is indicated in brackets under each category. *P*-values in the graphs are indicated as follows, *$p < 0.05$, ****$p < 0.0001$, and n.s.: non-significant. The number of cells is indicated in brackets in the figure legend.

---

environmental incubator box and a 40× C-Apochromat 1.2 NA water-immersion objective at 38.5 °C. Automatic 3D tracking was implemented for time-lapse imaging with a temporal resolution of 4 min using MyPiC[90]. mClover3 was excited with a 488 nm laser line and detected at 493 - 571 nm. mScarlet was excited with a 561 nm laser line and detected at 571–638 nm. SNAP-cell 647-SiR was excited with a 633 nm laser line and detected at 638–700 nm. Images of the control and experimental groups were acquired under identical imaging conditions on the same microscope. Some images shown have been filtered with a Gaussian filter of 1.3 in XY to reduce noise in ZEN (Zeiss).

Fixed samples were imaged in 2 µl of PBS with 10% BSA under paraffin oil in a 35 mm dish with a #1.0 coverslip (MATTEK, #P35G-1.0-1.4-C). Images were acquired using the Airyscan module on LSM800, LSM880 or LSM900 confocal laser scanning microscopes (Zeiss) and linear deconvolution was performed in ZEN (Zeiss). The acquisition was performed at room temperature with a 40× C-Apochromat 1.2 NA water-immersion objective. Imaging conditions were carefully selected (laser power, pixel-dwell time and detector gain) to avoid phototoxicity (for live imaging), photobleaching or saturation. For the analysis of kinetochore-microtubule attachments and telomere orientations, oocytes were manually rotated with an unbroken microinjection needle such that the long axis of the spindle was parallel to the imaging plane.

## STED microscopy

3D stimulation emission depletion microscopy (3D-STED) of fixed porcine oocytes was performed on an STED Expertline scanning microscope (Abberior Instruments GmbH) equipped with 405, 488, 561, and 640 nm excitation lasers and a pulsed 775 nm STED laser. Imaging was performed using a 100× oil immersion objective lens (N.A. 1.4, Olympus). Fixed oocytes, stained with Abberior STAR secondary antibodies were mounted in a drop of slow fade Glass Soft-set Antifade Mountant (Thermo Fisher Scientific; #S36817). STED Images were acquired (isotropically) with an isotropic 40 nm pixel size using the ADAPTIVE ILLUMINATION RESCUE function, to minimise laser exposure and reduce photobleaching. Two separate, sequential acquisitions have been performed for this experiment. The first acquisition was performed in fast confocal mode with a pixel size of 40 nm × 40 nm × 320 nm (XYZ). 4-colour imaging was performed of DNA, kinetochores, microtubules and the TALE. The second acquisition was performed by imaging the DNA and kinetochores in confocal mode and the kinetochores in STED mode using the RESCUE function with a pixel size of 40 nm × 40 nm × 40 nm (XYZ). The DNA and confocal kinetochore channels from both acquisitions were used to align the images in order to determine the angle of kinetochores to the spindle axis as described below in the analysis.

## Light-sheet microscopy

Data used for the bivalent tracking in Fig. 2a, c, Supplementary Fig. 2a–d most of the data for anaphase imaging in Figs. 2d–g and 1f, g and data for the chromosome location in the nucleus in Supplementary Fig. 2a were acquired on an LS1 Live light-sheet microscope system (Viventis) equipped with an environmental incubator box. A 25 × 1.1 NA Nikon water-dipping objective was used to collect the emitted fluorescence. Images were acquired with a light-sheet thickness of 1.6 µm. A customised script was used to track the sample in order to minimise the field of imaging and reduce phototoxicity.

## DNA fluorescence in situ hybridisation (DNA FISH)

Porcine oocytes arrested in metaphase II were fixed in 4% formaldehyde in phosphate-buffered saline (PBS) for 30 min at room temperature and subsequently extracted in 0.5% triton X (Sigma-Aldrich) in PBS (PBT) overnight. Oocytes were washed in PBT to remove any traces of formaldehyde and incubated in 0.1 M HCl for 10 min, before being washed in PBT again. FISH probes containing an ATTOTM 647N (NHS Ester) fluorophore at their 5′-end were purchased from Integrated DNA Technologies GmbH. FISH probes had the following sequences: Acrocentric forward: 5′-CCATGCAGCGTGATTGA-3′ and Acrocentric reverse: 5′-TCAATCACGCTGCATGG-3′; Metacentric forward: 5′-GCCTAGTTCTCACCTAGC-3′ and Metacentric reverse: 5′-GCTAGGTGAGAACTAGGC-3′. To detect either acro- or metacentric chromosomes, the respective forward and reverse oligos were each diluted to 100 nM in Formamide (99,3%, VWR Chemicals)/50% Dextran Sulfate solution (VWR Chemicals) (1:1) that additionally contained salmon sperm ssDNA (Abcam, AB229278) at 1 mg/ml. Oocytes were transferred from PBT into a drop of the diluted probes within a glass dish and covered with a glass coverslip. The dish was then placed on a 75 °C heating plate for 5 min before it was placed in a humidified box and incubated at 37 °C overnight. The dish was filled with PBT, cells were collected and washed again in PBT before being incubated in PBT

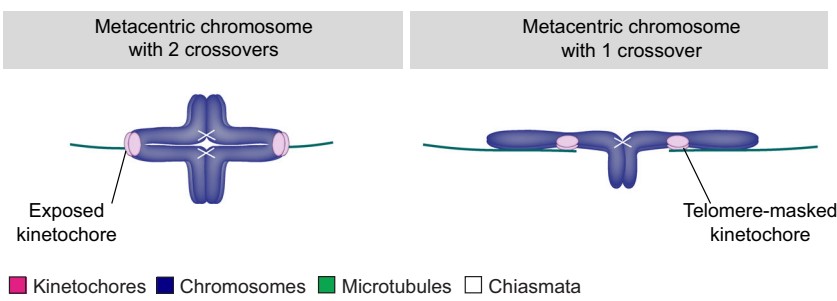

**Fig. 7 | Kinetochore masking and weak cohesion promote high aneuploidy of acrocentric chromosomes during mammalian meiosis. a** Scheme summarising how differences in acro- and metacentric chromosome bivalent morphology in meiosis I, and lower levels of pericentromeric cohesin in acrocentric chromosomes bias acrocentric chromosomes toward aneuploidy. **b** Model proposing an explanation for the necessity of metacentric chromosomes to form a chiasma on each of the two arms.

containing 100 μM Hoechst 33342 (Molecular Probes) for 10 min at room temperature. Oocytes were washed in PBT and subsequently imaged as described above.

### Data analysis

**Location of chromosomes in the GV nucleus.** Images from the GV nucleus of porcine oocytes acquired on the Viventis light-sheet microscope were analysed with Imaris 9.3. Oocytes were injected with both the acrocentric and the metacentric-TALE. The background signal of the TALE is weak within the nucleus in contrast to the cytoplasm. To segment the nucleus, the signal of the metacentric-TALE was inverted. A surface object was created on the inverted signal, and the nucleus' centre was computed. Spot objects were created on the acrocentric and the metacentric-TALE signal in the nucleus. Distances of the spots from the nucleus centre were computed using Matlab R2018b.

**Chromosome tracking.** Homologous kinetochores were tracked and paired before anaphase using an in-house plugin for Imaris (Bitplane). Mixed Integer Linear Programming (MILP) was used to find the optimal set of tracks and pairings simultaneously. Furthermore, MILP allows constraint introduction at the same time. All together this can reduce the manual corrections required compared to pairing after the tracks have been determined[91]. Tracked Filament objects in Imaris (Bitplane) were used to represent homologous kinetochores and their tracks. The script was written in Matlab R2018b and additionally used IBM Cplex with an academic license to improve performance.

Tracks were weighted by the square of the distance travelled between frames. The distance was determined by the mean distance travelled by both kinetochores. All distances are relative to a reference frame determined by the centre of mass of the chromosome signal.

Pairings were weighted by the chromosome signal on a line connecting the putative homologues. Furthermore, the angle between the line connecting the homologues and the spindle axis was minimised. The optimisation was run iteratively and the spindle axis was determined by the mean direction of the previously determined homologous chromosomes. Additional weights with little influence, are the variance of the chromosome signal between homologous chromosomes and a combination of the angle to the spindle axis with the interkinetochore distance to discourage far separated homologues from being non-parallel.

Constraints were introduced setting a maximum distance between homologues and a maximum distance travelled between frames. Line segments connecting homologues must be at least a certain distance apart and there should be 19 kinetochore pairs identified in each frame. Any missing kinetochore pair is heavily penalised. There must be at least one kinetochore somewhere near any detected spot. The chromosome signal was segmented and at least one kinetochore pair must be close to any connected component of this signal.

Depending on the number of spots and length of the time series, the script can take a few minutes to hours to finish. For this reason, time series of more than 100 frames are solved in smaller batches. The tracks can be easily stitched in Imaris. Manual corrections of tracks are performed in Imaris as well. Corrections of pairings are performed by deleting erroneous ones in Imaris and adding new ones via a separate script.

**Analysis of tracking.** An analysis script was written for Imaris in Matlab R2018b to obtain data on tracked kinetochore pairs from multiple files. The middle of the chromosome mass was determined after blurring and segmenting the chromosome mass via Otsu's method. The centre of mass was weighted by the chromosome intensity. The spindle axis was determined by the average direction of the lines connecting kinetochore pairs. More specifically, these lines are interpreted as 180-degree rotation vectors of length 1 and averaged using the Matlab function *meanrot*.

Positions of kinetochores and pairs were measured vertically, along the spindle axis from the metaphase plate. The metaphase plate is defined by the middle of the chromosome mass and the spindle axis as its normal vector. The absolute was taken for every distance.

**Large and small acrocentric chromosome group segmentation.** To determine the volume of the acrocentric chromosomes, they were segmented using an in-house script written in Matlab 2021b for Imaris 9.3. The default options were used. The *filaments* already used to determine the positions of the acrocentric and metacentric chromosomes were used as seeds to separate touching segments. This was repeated for many frames in the time series, and frames with poor segmentation were manually excluded.

Prior to segmentation, the image is interpolated along z, Gaussian smoothed and the background subtracted. Segmentation is based on Otsu's method (*multithresh*), but the number of thresholds is increased until a total chromatin volume in the desired range of 150–800 μm³ can be obtained by at least one. Any holes within the chromosomes are filled and small fragments at or below 20 interpolated voxels are removed. The segmentation is separated into smaller components using the standard watershed algorithm on the smoothed distance map of the segmentations. Typically, several components are generated per chromosome/sample. To recombine these, the total contact area between chromosomes is minimised. The minimisation is achieved by solving the multi-cut problem in the graph constructed from nodes representing the components and edges representing their contact areas. Integer linear programming (*MILP*) is used to solve this problem. Components with multiple seeds are separated between them using a distance map. Components that are disconnected from seeds in the graph are assigned by distance.

**Chromosome count in metaphase II.** Airyscan images were used for chromosome count analysis after Airyscan processing with the ZEN software. Analysis was done in Arivis Vision4D (version 3.1.1) software. The kinetochores were detected by automated thresholding of the kinetochore signal. The euploidy status of the egg was evaluated first independent of the presence of the TALE. An egg is considered euploid if 38 kinetochores are detectable and aneuploid if the number of kinetochores deviates from 38. Subsequently, the two kinetochores of a pair were manually assigned to each chromosome and annotated. The labelled kinetochores were identified by the presence or absence of the TALE in the kinetochore proximity. Acrocentric chromosomes were the ones with the TALE in proximity and Metacentric the ones without.

**Lagging chromosomes in anaphase.** Three colour imaging datasets of anaphase from either confocal or light-sheet microscopy were used. Images were acquired every 4 min. Anaphase onset was defined as the first time point where the chromosomes started separating into two anaphase masses. A small gap between the two masses is already visible at this stage. Chromosomes that are still present in the centre of the spindle 12 and 20 min after anaphase onset were classified as lagging chromosomes and were assigned as mildly and severely lagging, respectively. The mildly lagging group does not include the severely lagging chromosomes.

**Analysis of kinetochore-microtubule attachments.** Kinetochore-microtubule attachments were analysed in 3D high-resolution Airyscan images of the whole spindle volume in Imaris software version 9.3 (Bitplane). Only oocytes with the spindle oriented parallel to the imaging plane were used in this analysis, which provides the best resolution to assess the attachment. For all quantifications of human or porcine spindles, the kinetochore-microtubule attachments were assessed in the same way. An attachment is classified as end-on if the k-fibre stopped just in front of the kinetochore surface. An attachment

is classified as lateral if the k-fibre extended past the kinetochore and as merotelic if two distinct k-fibres originating from different poles interact with the kinetochore. A kinetochore is considered unattached if there is no interaction with microtubule filaments. All kinetochores were annotated in Imaris 2D view and then classified into one of the above categories. The data used for this analysis were two types of 4-colour images; oocytes stained for kinetochores, microtubules, TALE and DNA, and oocytes stained for kinetochores, microtubules, telomeres, and DNA. The quantification was done blindly for all kinetochores excluding the TALE and telomere channels from the analysis. Thus, the attachment type was assigned without knowing the chromosome category (acrocentric or metacentric) or the telomere configuration. In Fig. 5d, N/A refers to the amount of kinetochores that could not be quantified.

**Analysis of Mad1-positive kinetochores.** The number of Mad1-positive kinetochores was semi-automatically quantified in Imaris (version 9.3.0). Only injected oocytes that had their chromosomes aligned at the metaphase plate 12–13 h after release from RO-3306 were analysed. Analysis was done at one time point for each oocyte that corresponded at around 12 h after release. Based on the 568 channel, corresponding to Mad1, spots with an estimated XY diameter of 1.73 μm and an estimated $z$ diameter of 3.47 μm were created. A quality filter with the intensity mean of the 568 signal, corresponding to H2B, was used if required, to distinguish between Mad1 at kinetochores and background staining. Identified spots were then verified or rejected manually.

**Analysis of STED microscopy.** For the analysis, we used two image stacks acquired sequentially to preserve fluorescence. In stack1, we imaged, with isotropic sampling, the kinetochores (KTs) using STED and confocal microscopy, and DNA using only the confocal. In stack2, we imaged the kinetochores, TALE, microtubules (MTs) and DNA in confocal with a lower z-sampling. The analysis has been performed in Fiji[92] using different publicly available plugins and their own code.

Kinetochores in the STED image (stack1) were segmented semantically using CATS, a trainable pixel classifier in Fiji (Tischer & Pepperkok, 2019) followed by a connected component analysis with MorphoLibJ[93]. With Labkit[94] we manually inspected the label mask and if necessary corrected the segmentation. Typical segmentation errors were merged kinetochores and false positives. In 28/32 oocytes we correctly identified 38 kinetochores, in 4/32 we identified 34–37 kinetochores.

To define the position of the TALE signal we used the confocal images of the TALE and DNA channels from stack2. We wrote an ImageJ macro to perform instance segmentation of the bright TALE blobs within a mask defined by the DNA signal. Manual correction was done by inspecting TALE, the kinetochore and DNA channels with Labkit. Overall we were able to correctly detect 12 TALE objects in 15/32 oocytes. In 17/32 oocytes we identified 8–13 objects (median 10). To pair the kinetochores and TALE signals we first registered stack1 and stack2 by means of their respective confocal kinetochore signals using the ImageJ plugin *Correct 3D drift*[95]. The registration shifts are applied to the TALE centroids and used to compute the shortest distance to kinetochore centroids. Kinetochore centroids further away than 2 μm are considered non-TALE labelled.

The main axis of each kinetochore label was computed using MorphoLibJ and corresponds to the longest ellipsoid axis. Spindle direction was defined using the microtubule signal in stack2 and manually placed labels at each spindle pole. The absolute value of the dot-product between the kinetochore and spindle axis gives the kinetochore alignment to the spindle.

**Analysis of telomere position.** The position of telomeres with respect to the kinetochores was evaluated by 3D analysis of high-resolution Airyscan images of the whole spindle volume in Imaris version 9.3 (Bitplane). Only oocytes with the spindle oriented parallel to the imaging plane were used in this analysis. First, the kinetochores that had a telomere in their proximity were identified and annotated. Then, using the three channels of kinetochores, telomeres and DNA, the relative position of the telomere to the kinetochore was evaluated. To categorise the configurations, we considered the amount of contact between the telomeres and DNA, as well as the extent to which the telomeres covered the front part of the kinetochore. We classified kinetochores as either "telomere-masked" or "exposed" based on the visual centre of mass of the telomere. Telomere-masked kinetochores were those where the centre of mass of the telomere was shifted toward the spindle pole and where the telomere was located on the front surface of the kinetochores. In contrast, exposed kinetochores were those where the centre of mass of the telomeres was shifted toward the centre of the spindle and not covering the front part of the kinetochore.

**Measurement of MII sister interkinetochore distance.** The distance between the kinetochores of sister chromatids was measured using the measurement points feature in Imaris 9.3 (Bitplane). The data were then normalised to the mean interkinetochore distance of each cell.

**Measurements of SMC3 intensity MII.** We used an in-house script to measure the intensity of SMC3 in the pericentromeric region of sister chromatids. Sister kinetochores were paired using the measurement points feature in Imaris 9.3. (Bitplane). The SMC3 signal was measured in a round cylinder in the direction of the connecting filament between the two sister kinetochores. The cylinder radius was 0.5 μm. Background subtraction was performed, and data were normalised to the mean intensity of each cell.

All Imaris files within a folder were analysed and the results were recorded in an Excel sheet.

**Measurements of REC8 Intensity MI.** In order to measure the REC8 intensity in the pericentromeric region of MI bivalents, we wrote an in-house plugin for Imaris 9.3 (Bitplane) using Matlab R2018b. First, the kinetochores were identified using the spot object. The plugin changes the size of spots. This allows us to use the inbuilt spot function in Imaris to detect the centre of the signal of the kinetochore more easily and measure the mean REC8 intensity over a larger radius covering the pericentromeric region. The radius used for all kinetochores was 1 μm. Background subtraction was performed, and data were normalised to the mean intensity of each cell.

**Analysis of Trim-Away.** Oocytes were co-labelled for DNA and the acrocentric and metacentric-TALE within the same cell. Analysis was performed after 3D reconstruction in Imaris 9.3 (Bitplane). The time when two sister chromatids were separated from each other was noted for all the labelled chromosomes. Time point zero in the graph marks the time of the first separation event for each egg.

**Segmentation for the 3D chromosome movies.** The 3D rotating chromosome movies were created using the animation feature of Imaris 10.0. Imaris surfaces around chromosomes, kinetochores and telomeres were manually cut and merged. They were then dilated by 0.25 μm using Imaris' built-in distance transformation. Further manual cuts were made to remove any remaining elements from neighbouring chromosomes. All visible channels were masked using the resulting surfaces. The movie is maximum intensity projected and interpolated by Imaris with an orthogonal perspective.

**Statistical analysis.** Statistical significance based on unpaired, two-tailed Student's *t*-test (for absolute values) was calculated in OriginPro and two-sided Fisher's exact test (for categorical values) in Prism (GraphPad). No adjustments were made for multiple comparisons. All

box plots show median (horizontal black line), 25th and 75th percentiles (boxes) and outliers or SD (whiskers). All data are from at least three independent experiments apart from the cold-stable assay in metaphase II of pig oocytes and the FISH labelling which are from two independent repetitions. $P$-values are designated as $*P < 0.05$, $**P < 0.01$, $***P < 0.001$ and $****P < 0.0001$. Non-significant values are indicated as "n.s.".

## Reporting summary

Further information on research design is available in the Nature Portfolio Reporting Summary linked to this article.

## Data availability

Plasmids are available from M.S. under a material transfer agreement with the Max Planck Society. The datasets generated and analysed as part of the current study are available from the corresponding author on request. Due to their large size, the primary microscopy data were not uploaded to a data repository but are available from the corresponding author on request. Source data are provided with this paper.

## Code availability

Scripts and plugins are available at: https://doi.org/10.17617/3.NPIDK2.

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

## Acknowledgements

We are grateful to the patients who participated in this study. We thank the staff from the Facility for Light Microscopy at the Max Planck Institute for Multidisciplinary Sciences for technical assistance; the clinicians, nursing team, and embryology team at the Fertility Center Berlin for their support of this study; C. So, and L. Wartosch for help with human oocytes; T. Cavazza, C. So, and K. Harasimov for help to optimise the pig oocyte culture system; M. Daniel for help with the ovaries delivery; C. So and P. Lenart for helpful discussions; C. So, and Life Science Editors for critical comments on the manuscript; S. Cheng, J. Ellenberg, A. Webster and K. Harasimov for plasmids and A. Webster for antibody. The research leading to these results was funded by the Max Planck Society and the DFG under a Leibniz Prize to M.S. (SCHU 3047/1-1).

## Author contributions

M.S. conceived the study. E.B. and M.S. designed experiments and methods for data analysis. A.P.Z. designed the acrocentric and meta-centric TALEs, cloned the acrocentric-TALE, recorded the first confocal images of chromosomes with the acrocentric-TALE in porcine oocytes, and established the initial protocol for porcine oocyte culture and imaging. Confocal imaging and oocyte culture conditions were then further developed by E.B. Light sheet imaging was established by E.B. N.S performed the FISH experiments for Supplementary Fig. 1d, helped with the Mad1 experiments for Supplementary Fig. 3a, b and tested outer kinetochore antibodies. A.W. performed the fixation of metaphase II oocytes for Fig. 1d, e. E.B. performed all other experiments. E.M. wrote the bivalent tracking script used for Fig. 2c and Supplementary Fig. 2d and the rest of the in-house-developed Matlab scripts used for Supplementary Fig. 2a–c and Fig. 6d, f. A.Z.P. designed methods and scripts for STED image analysis in Fig. 4f–j. E.B. performed all analyses with the following exceptions: A.Z.P. analysed the data from STED microscopy in Fig. 4f–j. E.M. annotated the kinetochores for tracking for Fig. 2c and Supplementary Fig. 2d. N.S. performed the analysis of Supplementary Fig. 3a. A.W. annotated the kinetochores in the fixed metaphase II oocytes used in Fig. 1d, e. C.S. and A.T.S. supervised the work in the Berlin Fertility Center. E.B. and M.S. wrote the manuscript and prepared the figures with input from all authors. M.S. supervised the study.

## Funding

## Competing interests

The authors declare no competing interests.
