## [Transparent Peer Review file · Nature Communications]

Chromosome architecture and low cohesion bias acrocentric chromosomes towards aneuploidy during mammalian meiosis

Corresponding Author: Professor Melina Schuh

Version 0:

Reviewer comments:

Reviewer #1

(Remarks to the Author)

The authors have addressed all of my comments. The following are additional, very minor comments for further improvement.

Minor comments:

1. They can consider adding a statistical test for Fig. 2C.
2. Fig. S2A - The color for the Acro-TALE box is too dark to see the median line.

Reviewer #3

(Remarks to the Author)

The authors have done an excellent job of responding to the reviewers' comments. In particular, the toning down of language to avoid over-stating the observations is greatly appreciated. Although this manuscript is largely observational and the cause of poor segregation of acrocentric chromosomes cannot be established, it is very valuable for several reasons. First, the authors clearly demonstrate using live imaging that acrocentric chromosomes are prone to mis-segregation. Second, though correlative, the authors provide two potential explanations for the lower fidelity segregation of acrocentric chromosomes: kinetochore masking and reduced centromeric cohesin. Third, the manuscript is a technical tour-de-force – the chromosome labelling strategy is powerful and important. I support publication in Nature Communications though there are a few points that should be addressed:

1. Figure 3C and Supplementary Figure 4A. It is very difficult to see the difference between telomere-masked and exposed kinetochores in the examples given. What was the criteria used by the authors to distinguish between these classes as some of the examples given look very similar despite being defined by the authors as being in different classes. For example, in Figure 3C, the examples given for unattached look extremely similar, despite the authors categorising the top image as "Telomere masked" and the bottom image as "exposed kinetochore". Is the difference the DNA staining (which is difficult to see)? But if that is the case, the "End on" examples do not make sense as there is no DNA stain on the microtubule side of the top image which is categorised as "Telomere masked". The categorisation here seems rather subjective and it would be better if the authors could use a software-based method to define the classes, rather than human judgement which is subject to unconscious bias.
2. Figure 6D which comparisons are "ns"? This is not clear from the way the bracket is drawn. Can the authors add a more detailed explanation to the figure legend.
3. Line 233 "Only" here seems a bit of an overstatement and is suggested to be deleted.

Version 1:

Reviewer comments:

Reviewer #3

(Remarks to the Author)

The authors have satisfactorily addressed all outstanding comments of the reviewers and is suitable for publication.

Reviewer #1 (Remarks to the Author):

The authors have addressed all of my comments. The following are additional, very minor comments for further improvement.

We thank the reviewer for the very positive comments and for supporting the publication of our manuscript in Nature Communications. We have addressed the two very minor comments as detailed below.

Minor comments:

1. They can consider adding a statistical test for Fig. 2C.

We thank the reviewer for the helpful suggestion. Graph 2C shows the average distance of all acrocentric and metacentric chromosomes from the metaphase plate. To be able to perform a statistical test, we have created an additional graph where we plotted the mean distance of each acrocentric and metacentric chromosome from the metaphase plate over time. Consistent with our previous quantification, acrocentric chromosomes were significantly more distant from the metaphase plate than metacentric chromosomes (new Supplementary graph 2C).

2. Fig. S2A - The color for the Acro-TALE box is too dark to see the median line.

We thank the reviewer for highlighting this point. We have now used a lighter shade of grey that is more distinct from the colour of the median line (new graph below).

Reviewer #3 (Remarks to the Author):

The authors have done an excellent job of responding to the reviewers' comments. In particular, the toning down of language to avoid over-stating the observations is greatly appreciated. Although this manuscript is largely observational and the cause of poor segregation of acrocentric chromosomes cannot be established, it is very valuable for several reasons. First, the authors clearly demonstrate using live imaging that acrocentric chromosomes are prone to mis-segregation. Second, though

correlative, the authors provide two potential explanations for the lower fidelity segregation of acrocentric chromosomes: kinetochore masking and reduced centromeric cohesin. Third, the manuscript is a technical tour-de-force – the chromosome labelling strategy is powerful and important. I support publication in Nature Communications though there are a few points that should be addressed:

We thank the reviewer for the very positive feedback on the revised version of our manuscript and are pleased that the reviewer now supports publication in Nature Communications. We have addressed the remaining few points as detailed below.

1. Figure 3C and Supplementary Figure 4A. It is very difficult to see the difference between telomere-masked and exposed kinetochores in the examples given. What was the criteria used by the authors to distinguish between these classes as some of the examples given look very similar despite being defined by the authors as being in different classes. For example, in Figure 3C, the examples given for unattached look extremely similar, despite the authors categorising the top image as “Telomere masked” and the bottom image as “exposed kinetochore”. Is the difference the DNA staining (which is difficult to see)? But if that is the case, the “End on” examples do not make sense as there is no DNA stain on the microtubule side of the top image which is categorised as “Telomere masked”. The categorisation here seems rather subjective and it would be better if the authors could use a software-based method to define the classes, rather than human judgement which is subject to unconscious bias.

We acknowledge that a software-based automated analysis would be ideal. However, despite our attempts, automated approaches have proven unreliable due to the complex 3D arrangement of chromosomes, their telomeres, and microtubule attachments. An automated analysis would require consideration of multiple factors that are difficult to assess automatically, including the orientation of the spindle axis, the position and shape of chromosomes, and the detection and allocation of telomeres to specific kinetochores. An automated analysis is further complicated by the fact that two telomeres sometimes appear as a single spot, making automated detection even more difficult. Given these challenges, we have developed a qualitative assessment method based on a well-defined workflow. We have now provided a more detailed description of this process in the methods section, along with an illustrative figure that demonstrates how different configurations affect the accessible kinetochore surface.

In our method, we used telomere staining as a proxy for visualizing the small arm of chromosomes. In acrocentric chromosomes, the two telomeres are in close proximity to the kinetochores. To categorize the configurations, we considered the amount of contact between the telomeres and DNA, as well as the extent to which the telomeres covered the front part of the kinetochore. We classified kinetochores as either “telomere-masked” or “exposed” based on the visual centre of mass of the telomere. Telomere-masked kinetochores were those where the centre of mass of the telomere was shifted towards the spindle pole and where the telomere was located on the front surface of the kinetochores. In contrast, exposed kinetochores were those where the centre of mass of the telomeres was shifted towards the centre of the spindle and not covering the front part of the kinetochore.

These configurations significantly affect the accessible surface of the kinetochore. To make this concept clearer, we have added a new illustration (Supplementary Figure 4C).

In the example of unattached kinetochores in Figure 4C (we believe the reviewer meant 4C instead of 3C), the telomere-masked example shows one telomere in proximity to the DNA and another not touching the DNA. In contrast, the exposed example displays both telomeres in proximity to the DNA. Additionally, in the telomere-masked example, the telomere covers the front part of the kinetochore. Together, these configurations lead to major differences in accessible kinetochore surfaces, as now also illustrated in Supplementary Fig. 4C.

2. Figure 6D which comparisons are “ns”? This is not clear from the way the bracket is drawn. Can the authors add a more detailed explanation to the figure legend.

We thank the reviewer for the helpful comment. We have now separated both brackets by a gap for clarity. In addition to Fig. 6D, we applied the same changes to Fig. 6B and 6F.

3. Line 233 “Only” here seems a bit of an overstatement and is suggested to be deleted.

We thank the reviewer for the helpful comment, and have removed the “only” from the revised version of this manuscript.

Comments related to our response to Reviewer 2:

1. Regarding reviewer #2 point t#5, reviewer #3 mentioned that this relates to point #1 in their own report and noted that they were not entirely convinced by the claim of ‘masking’ or that it had a role in the mis-segregation of acrocentrics. Reviewer #3 was of the opinion that further toning down on this point would be advisable.

We thank the reviewer for the helpful comment. We have modified the wording throughout the manuscript to emphasize that there is a correlation between telomere-masked kinetochores and incorrect kinetochore-microtubule attachments, and we no longer refer to a causal relationship. We have now gone through the manuscript again to validate that all positions have been changed and modified the wording in a few additional places to be on the very safe side.

2. Regarding reviewer #2 point #9, reviewer #3 advised that because there is a lot of variability in the timing of human oocyte progression it is impossible to say definitively which stage they are in without other markers. Reviewer #3 advised that you state in the results that it cannot be ruled out that the oocytes were in different stages of prometaphase/metaphase where the attachments have not matured.

We thank the reviewer for this helpful comment. We have now included the following sentence in the results section of our revised manuscript: “All human oocytes were monitored for chromosome

alignment and fixed at around 15h30min post NEBD. However, we cannot rule out that individual oocytes might have been at slightly different stages of development at the time of fixation.”